# Deep Learning Model Based on You Only Look Once Algorithm for Detection and Visualization of Fracture Areas in Three-Dimensional Skeletal Images

**DOI:** 10.3390/diagnostics14010011

**Published:** 2023-12-20

**Authors:** Young-Dae Jeon, Min-Jun Kang, Sung-Uk Kuh, Ha-Yeong Cha, Moo-Sub Kim, Ju-Yeon You, Hyeon-Joo Kim, Seung-Han Shin, Yang-Guk Chung, Do-Kun Yoon

**Affiliations:** 1Department of Orthopedic Surgery, University of Ulsan College of Medicine, Ulsan University Hospital, Ulsan 44033, Republic of Korea; 2Department of Integrative Medicine, College of Medicine, Yonsei University of Korea, Seoul 03722, Republic of Korea; back582cool@naver.com; 3Industrial R&D Center, KAVILAB Co., Ltd., Seoul 06675, Republic of Korea; hazero@kavilab.ai (H.-Y.C.); mskim@kavilab.ai (M.-S.K.); youstar@kavilab.ai (J.-Y.Y.); hyeonjookim@kavilab.ai (H.-J.K.); 4Department of Orthopedic Surgery, Seoul St. Mary’s Hospital, College of Medicine, The Catholic University of Korea, Seoul 06591, Republic of Korea; tumorshin@gmail.com (S.-H.S.); ygchung@catholic.ac.kr (Y.-G.C.)

**Keywords:** YOLO v4, fracture detection, deep learning, three dimensional (3D) reconstructed image, tibia and elbow

## Abstract

Utilizing “You only look once” (YOLO) v4 AI offers valuable support in fracture detection and diagnostic decision-making. The purpose of this study was to help doctors to detect and diagnose fractures more accurately and intuitively, with fewer errors. The data accepted into the backbone are diversified through CSPDarkNet-53. Feature maps are extracted using Spatial Pyramid Pooling and a Path Aggregation Network in the neck part. The head part aggregates and generates the final output. All bounding boxes by the YOLO v4 are mapped onto the 3D reconstructed bone images after being resized to match the same region as shown in the 2D CT images. The YOLO v4-based AI model was evaluated through precision–recall (PR) curves and the intersection over union (IoU). Our proposed system facilitated an intuitive display of the fractured area through a distinctive red mask overlaid on the 3D reconstructed bone images. The high average precision values (>0.60) were reported as 0.71 and 0.81 from the PR curves of the tibia and elbow, respectively. The IoU values were calculated as 0.6327 (tibia) and 0.6638 (elbow). When utilized by orthopedic surgeons in real clinical scenarios, this AI-powered 3D diagnosis support system could enable a quick and accurate trauma diagnosis.

## 1. Introduction

From a medical engineering perspective, “artificial intelligence (AI)”-based diagnostic technology has developed greatly. AI has been applied to various platforms from large medical devices to portable mobile devices. For example, diagnoses using AI are used in various places in the medical field, such as AI models to check the bone density status or predict fracture risk from medical images [1]. According to the most recent research, there have been many studies on the diagnosis of fractures from X-ray images using AI. Kuo et al. demonstrated the usefulness of AI for predicting fractures from radiation images [2]. Kolanu et al. present a study on diagnosing fractures using the “X-Ray Artificial Intelligence Tool (XRAIT)” [3]. All of the above cases are methods for the diagnosis of bone fractures from X-ray images with a high accuracy. However, most studies only showed the performances of AI for diagnosing bone fractures using two dimensional (2D) images. Naturally, the 2D images included a lot of information which can provide a basis for the diagnosis of a bone fracture. However, some factors that could reveal the characteristics of the bone fracture may be lost due to dimensional limitations. Moreover, because the overlapped object in 2D X-ray images can disturb the identification of the affected region, an intuitive diagnosis can be made using 3D images instead of 2D images. In particular, identifying the impacted region within an image of a comminuted fracture has been challenging. An inaccurate diagnosis in the case of a bone fracture can create critical problems for patients, and it can be one of the major factors that lengthens the operation time [4].

However, AI can assist in the detection of the fracture region and to make a decision for the diagnosis and surgical strategy. The diagnosis of bone fractures is classified as an object detection task in the field of computer vision. There are many deep learning models for object detection in medical images. And, a dedicated dataset can make the model show better performance for the object detection task. Because the fracture detection in the medical images is difficult, models which can show an excellent detection performance are essential. The representative model for object detection is the “you only look once (YOLO)” model. The first YOLO model (YOLO v1) was presented in 2016 by Joseph et al. [5,6]. The latest model is known as YOLO v8 which was presented in 2023. YOLO v4 was developed by Alexey et al. in 2020. YOLO v4 was constructed to train the model using low-level hardware conditions (e.g., GTX 2080 Ti). Moreover, it uses the Bag of Freebies and the Bag of Specials methods with spatial pyramid pooling (SPP) and a path aggregation network (PAN) [7,8]. In addition, YOLO v4 was developed to become YOLO v7 which can detect objects in real time [9,10].

As an example, a research endeavor employed YOLO v4 to proficiently detect lung nodules [11]. The findings demonstrated the effectiveness of YOLO v4 within a hospital hardware environment. In one of the more recent investigations, object recognition using YOLO v4 was scrutinized, revealing substantial enhancements in both detection speed and accuracy. Notably, a recent study harnessed YOLO v4 to explore the classification and detection capabilities using extensive medical data, and demonstrated a commendable performance [12]. The ongoing refinement of YOLO v4’s performance is evident, with continuous updates and user-driven feedback contributing to its improvement. Through these investigations, it can be discerned that YOLO v4’s performance has experienced consistent advancement, underscoring the ongoing research endeavors in the realm of object-based detection. And, YOLO v4 is one of the proper models that can detect bone fractures using medical images [13]. YOLO v4 can detected the fractured region using a bounding box on 2D computed tomography (CT) images, one slide at a time; bounding boxes in several image slides can show an intuitive fractured region in the 3D reconstructed image for the bone region. In this case, a more accurate diagnosis can be obtained, and errors in diagnosis will be reduced before the surgeons begin the operation. Moreover, leveraging YOLO v4, the system can intuitively guide diagnoses, irrespective of the surgeon’s skill level and experience, further enhancing its efficacy [10]. The purpose of this study was to evaluate the performance of the YOLO v4-based fracture detection method, and to show the possibility of intuitive diagnoses through a combination of 3D image reconstruction and fracture detection. In 3D space, the precise location of the wound can be identified, allowing for the accurate identification of fragmented or cracked bones [14]. This study is different from previous studies and significant in that it detects the fracture region more intuitively and accurately through 3D visualization [15,16]. To date, no studies have formulated a model capable of automatically reconstructing and identifying the fracture site in three dimensions. The imperative nature of such surgeries underscores the significance of this study.

## 2. Materials and Methods

### 2.1. Overall Workflow and Data Pre-Processing

The methods in this study were conducted according to the relevant guidelines and regulations of the Clinical Trial System of the Catholic Medical Center (CTSC). The experimental protocols in this study were approved by the Institutional Review Board (IRB) at Seoul St. Mary’s Hospital, the Catholic University of Korea (approval number: KC20RISI1034). The data pre-processing was performed by using MATLAB (2022b, Mathworks, Natick, MA, USA), which is software optimized for analyzing experimental data and processing data [17]. There are five steps for pre-processing the data in this study. Figure 1 shows the work flow for the overall process in this study [18]. MATLAB offers numerous supplementary modules tailored for researchers, providing the benefit of expediting model design. This acceleration is achieved by leveraging pre-implemented functions for preprocessing, training, validation, and testing. MATLAB holds considerable prevalence in medical research and is a robust tool, as demonstrated by its application in recent drug-related studies leading to FDA approval [19].

The first step for data pre-processing was the preparation of the data. The format of all the data was the CT Digital Imaging and Communications in Medicine (DICOM) image type; the CT images for the 155 cases included cases of comminuted fractures for several regions such as the tibia and elbow. In total, 250,000 individual CT images were used for training via data augmentation. The data augmentation was conducted 10 times using random shifts along the X and Y axes. The second step is the upload of the CT DICOM files to the image labeler of MATLAB. After uploading the DICOM files, the bounding boxes were manually added on the fracture region of the CT image, one by one (the third step). The bounding box has four kinds of information to demonstrate the right position on the CT image: X-coordinate of the starting point for the bounding box, Y-coordinate of the starting point for the bounding box, the number of pixels on the X-axis for the horizontal line, and the number of pixels on the Y-axis for the vertical line. All the information of the bounding boxes were stored by in format of a 4 by N matrix (the fourth step). It was used for as training images to train the network. Lastly, the DICOM images were converted into the Portable Network Graphics (PNG) format.

The model was constructed based on the YOLO v4 architecture, and training involved feeding preprocessed datasets for both the tibia and elbow into the designed model. Following training, the model underwent testing and validation using separately curated datasets for the tibia and elbow. The evaluation of the 3D reconstructed bone and fracture bounding box, obtained during testing and validation, encompassed metrics such as the loss function, Precision–Recall Curve (P-R curve), and Intersection over Union (IoU). Subsequent to this evaluation, iterations of additional data preprocessing or model optimization were conducted based on the assessment results.

In the dataset construction phase, the data underwent categorization based on the anatomical regions of the tibia and elbow. The categorized dataset underwent preprocessing, transforming it into a training dataset. Post MATLAB-based preprocessing for fracture site identification, the data were organized into datasets corresponding to the tibia or elbow region. Within the dataset, each entry encapsulated details about the fracture sites’ location, size, and label names, which were assigned during the preprocessing phase. Out of the complete dataset, 60% was allocated for training, 10% for validation, and the remaining 30% for testing. Each dataset earmarked for training, validation, and testing incorporated critical information concerning the location and size of the fracture labels, which are integral for the mapping of fracture sites onto masks in the 3D reconstructed bone images.

### 2.2. Modeling and Training

The construction of the deep learning models was performed using MATLAB. The hardware for the training models used two graphics processing units (GPUs) as NVIDIA GeForce RTX 3090 with 24 GB GPU memory, and 2.10 GHz dual Intel(R) Xeon(R) Silver processors with 128 GB RAM (Intel, Santa Clara, CA, USA).

The initial learning rate was established at 0.001, coupled with a predefined set of drop periods affecting the learning rate. The drop period, denoting the epoch interval for reducing the learning rate, was fixed at 20, accompanied by a 0.5 drop factor. Despite the initial maximum epoch being set at 50, the minibatch size was configured at 4. The chosen learning approach involved iterative training utilizing adaptive moment estimation (ADAM). The structure of YOLO v4 consists of a “Input”, “Backbone”, “Neck”, and “Head”. The overall structure of YOLO v4 can be seen in Figure 2. The Input part accepts a PNG file transformed from a DICOM file. The data flowing into the backbone of the cross stage partial connection DarkNet-53 (CSPDarkNet-53) is augmented. Data augmentation primarily involved implementing 2D affine transformations on both the labeled bounding box and CT images. The images underwent X-axis flipping or random scaling transformations ranging from 1 to 1.3 times. The decision to exclude Y-axis flipping stems from the regular shape of medical images, eliminating the necessity for such transformations. In contrast, flipping along the X-axis was adopted to enhance the learning experience related to the tibia or elbow, aligning with the right and left perspectives. Serving as both a convolutional neural network and a backbone for object detection, CSPDarknet-53 employs a split and merge strategy, promoting an increased gradient flow within the network [20]. The CSPDarkNet-53 model in Figure 3 is a model that dramatically reduces the amount of computation by combining the cross-stage hierarchy method of the layer based on DenseNet [21]. There are multiple ‘CSP Blocks’ which meticulously structured to allow for flexible output sizing (Figure 3a). The initial CSP Block comprises a singular layer, yielding an output dimension of 256 × 256 × 128. The second CSP Block incorporates two layers, resulting in an output size of 128 × 128 × 128. The third CSP Block generates an output size of 64 × 64 × 256. Likewise, the fourth CSP Block produces an output of 32 × 32 × 512. Concluding the sequence, the fifth CSP Block, comprising four layers, delivers an output of 16 × 16 × 1024. Within the confines of the CSP block, convolution is succeeded by batch normalization and activation through the mish function, contributing to the reduction of the loss value (Figure 3b). The Input part goes through the Backbone network and is transformed into a feature map with semantics. The Neck part consists of Spatial Pyramid Pooling (SPP) and a Path Aggregation Network (PAN). The SPP divides the last feature map of the convolution layer into a fixed-size grid and averages the values from the feature map to obtain a fixed-size representation, and PAN uses bottom-up path augmentation to easily pass low-level features to high-level features for more accurate localization [22,23]. In particular, the Neck part connects the Backbone and Head, and refines and reconfigures the feature map. The Head part processes the aggregated features using a one-stage detector method and predicts the bounding box, object score, and classification score.

The Bag of Freebies constitutes an amalgamation of diverse training techniques designed to enhance the accuracy of object detectors without incurring additional inference costs [24,25]. This methodology represents an offline training approach aimed at elevating the overall accuracy without a corresponding increase in overall inference costs. Conversely, the Bag of Specials encompasses an assortment of plugins and pre-processing modules that marginally augment the inference cost but can also substantially enhance the accuracy of the object detection [26]. The utilization of plugin modules and pre-processing to slightly elevate inference costs while significantly boosting accuracy is referred to as the Bag of Specials method. Typically, these plugins are engineered to augment specific attributes, such as expanding the receptive field, introducing attention mechanisms, or fortifying feature integration capabilities.

### 2.3. Evaluation and Analysis

The YOLO v4-based AI model was evaluated through “Precision-Recall curve (PR curve)”. The PR curve follows the basic attributes of classification evaluation metrics, and there are four basic attributes of classification evaluation metrics: true positive (TP), false positive (FP), true negative (TN), and false negative (FN). The “TP” attribute indicates that the AI model successfully predicted a positive classification, the “TN” attribute indicates that the AI model successfully predicted a negative classification, and the “FP” attribute indicates that the AI model predicted a positive classification which was actually negative, and the “FN” attribute indicates that the AI model predicted a negative classification which was actually positive. Precision is the percentage of “positive” predictions made by the AI model that are actually true. The precision was calculated as follows (Equation (1)):(1)$$\left(Precision\right)=\frac{true\;positive\;(TP)}{true\;positive\;\left(TP\right)+false\;positive\;(FP)}$$

Recall is the percentage of predictions that are actually true and also have a “positive” prediction. The recall was calculated as follows (Equation (2)):(2)$$\left(Recall\right)=\frac{true\;positive\;(TP)}{true\;positive\;\left(TP\right)+false\;negative\;(FN)}$$

Although precision and recall are both ratios of how well the AI obtains the correct answer, precision is focused on lowering the value of FP and recall is focused on lowering the value of FN. According to the PR curve, a good AI model should have high precision and recall values. In other words, a PR curve is a graph for evaluating the performance of a model that shows the relationship between the AI’s precision and recall. To show the correlation between the precision and recall of the YOLO v4 model using a PR curve, the YOLO v4 model was used to predict the tibia and elbow fracture sites that the YOLO v4 model had been previously trained with.

The IoU serves as a metric that indicates the success of the detection for an individual object. The derivation of the IoU is as follows (Equation (3)):(3)$$\left(intersection\;over\;Union\right)=\frac{Area\;of\;Intersection}{Area\;of\;Union}$$

The term “Area of Intersection” refers to the region corresponding to the correct answer, while “Area of Union” pertains to the predicted area. The IoU can achieve a maximum value of 1, with a higher value signifying a more precise prediction [27].

### 2.4. Three-Dimensional Image Reconstruction and Plotting Mask

Initially, details such as the width and height, slice thickness, and resolution of the DICOM image were extracted from the header of the DICOM file [28,29]. Given that the 3D conversion of DICOM images necessitates exclusively retrieving bone-related image information, a blank matrix was populated with 1 based on the origin coordinate axis, aligning with the Threshold set derived from the Hounsfield Unit (HU) range of 1150–1250. Subsequently, the populated matrix was transformed into a representation isolating only the bone segments from a single DICOM file. Equivalent curves were then derived from the finalized binary matrix to construct the bone surface.

To envelop the outer epidermis of the bone, composed of dots and lines, a layer of cotton was applied, with the upper and lower sections capped to fashion a complete bone structure. For the introduction of bounding boxes onto the 3D reconstructed bones, the DICOM image’s pixel information in the DICOM header was translated into distances along the X-axis and Y-axis. Calculating the starting point of the bounding box from the origin, the bounding box was subsequently mapped from this starting point onto the X-axis, Y-axis, and thickness, strategically placed within the fracture area.

Figure 4 is a diagram that briefly shows the process of positioning the bounding box onto the bone reconstructed in 3D. The DICOM header stores pixel information, height and width of the image, and threshold information of the HU range for the bone. In order to create a bounding box, pixel information from the DICOM file was converted into a 3D distance, and the starting position of the bounding box was drawn based on the coordinate axis. The thickness of the bounding box was constructed using the information on thickness in the DICOM file. In Figure 4, the red squares are the bounding boxes located in the fracture area.

## 3. Results

### 3.1. Performance of Model

Figure 5 displays the outcome of the validation loss post-training. On the Y-axis and X-axis, the loss and iteration are depicted, signifying the count of validations for the compact datasets. Within the curve, the blue line illustrates the fluctuation in loss during training, while the black line represents the actual validation loss variation. When analyzing the loss fluctuation based on iteration using CT images featuring tibia region fractures, the ultimate validation loss amounted to 4.87. Similarly, examining the loss variation after validation based on iteration using CT images featuring elbow region fractures yielded a final validation loss of 3.90.

Figure 6 illustrates sample outcomes depicting the detection of fractured areas utilizing YOLO v4 and 3D reconstructed images. The left segment displays the outcomes related to a tibia case, while the right part exhibits the findings concerning an elbow case. The initial image showcases the 2D representations of both the tibia and elbow, highlighting the fractured regions. The second image showcases the fracture site predictions made by YOLO v4. The third image exhibits the 3D reconstructed images derived from the CT series, incorporating the CT image from the first illustration. The fourth image presents the 3D reconstructed images, featuring a red mask effectively highlighting the fractured regions.

In Figure 7, the representative results are presented to demonstrate the detection performance for fractured regions in the tibia, including the fibula, from CT images. The entity detection network YOLO v4 delineated the fracture site on the CT images using a yellow bounding box. This bounding box not only pinpoints the exact location of the fractured area but it also provides a score based on the probability of an accurate detection.

Figure 8 exhibits representative results indicating the detection performance for fractured regions in the elbow (humerus, radius, and ulna) in CT images. The various yellow bounding boxes specify the fractured areas in the elbow region in the CT images. The creation of bounding boxes varied according to the size of the fracture. In instances of comminuted fractures affecting the entire structure, as visible in the CT image, the bounding boxes covered a wider range, indicating the extensive impact of the fracture.

### 3.2. Three-Dimensional Image Reconstruction and Visualization of Fractured Area

Figure 9 shows representative results featuring 3D reconstructed images of fractured tibia and fibula bones, with red masks indicating the fractured regions. These images were generated from a CT series, comprising multiple CT slides. YOLO v4 successfully detected the fractured regions, specified by the bounding boxes. Although these bounding boxes were essentially 2D representations on individual CT images, the 3D depiction (red mask) was achieved by amalgamating multiple bounding boxes from various CT images within a series. By stacking the bounding boxes in multiple layers, the fracture area could be intuitively identified.

Figure 10 illustrates representative results related to the elbow (humerus, radius, and ulna), which are 3D reconstructed images of fractured bones with red masks indicating the specific fractured regions. Given the elbow’s complex structure involving multiple bone fragments, the red mask provides an intuitive representation of the fractured area in the 3D reconstructed image. All regions affected by the fracture were prominently marked with the red mask.

### 3.3. Evaluation and Analysis

Figure 11 shows two Precision–Recall (PR) curves, illustrating the precision and recall of the fractured region detection facilitated by YOLO v4. The Y-axis delineates precision, while the X-axis represents recall. In Figure 10a,b, these PR curves were generated utilizing distinct datasets tailored for the tibia and elbow, respectively. Notably, these curves revealed average precisions of 0.71 and 0.81 in Figure 10a and Figure 10b, respectively, providing a comprehensive overview of the detection accuracy achieved by the YOLO v4 model.

In the case of the tibia, the IoU yielded a value of 0.6327, while for the elbow, the IoU registered at 0.6638. The standard deviation of the IoU of the elbow was 0.2709, and 0.2754 for the tibia. Notably, the IoU demonstrated relatively high values despite the limited availability of learning data. Typically, when the IoU reaches 0.5 or above, the model is considered to exhibit compliant performance. This is because an IoU surpassing 0.5 implies effective coverage of the ground truth area, equivalent to two-thirds of the region [30,31].

## 4. Discussion

In this study, we demonstrated the accurate visualization of fractured regions using deep learning and 3D reconstructed images for the fractured bone from CT images. In other words, the purpose of this study was to evaluate the performance of the YOLO v4-based fracture detection method, and to show the possibility of an intuitive diagnosis by combining 3D image reconstruction and fracture detection [32]. The possibility of finding the fractured region from the 2D/3D CT images can be decreased according to internal or external conditions. In particular, there are several limitations in 2D images, such as low contrast, a lot of noise, etc. In the case of 3D reconstructed CT images, although most cases provide better visualization than the 2D images, there are still difficulties in detecting the fractured regions due to the following reasons: over-processing to smooth mesh, occluded objects, unclear fracture lines, etc. For these reasons, there are always factors affecting the accurate diagnosis of orthopedic trauma from 2D/3D images. In this study, the methodology to generate a 3D reconstructed bone image with automatic detection of the fractured regions which can be easy to miss (such as Figure 8 and Figure 9) was suggested to make up for the weakness of the conventional method for the diagnosis of orthopedic trauma. Additionally, it is imperative to conduct more nuanced comparative studies to determine the necessity of utilizing YOLO v4. Despite the proliferation of YOLO model versions, YOLO v4 has demonstrated notable efficacy and accuracy even in resource-constrained environments. A noteworthy observation is that YOLO v4 not only exhibited compliant performance but also has credibility due to the availability of relevant research papers. When opting for a model, careful consideration should be given to selecting the most suitable one based on its accuracy and the specific context of its application. The accompanying table provides a reference for the requisite conditions related to performance and utilization [33].

We explored the latest methodologies for diagnosing fractures, uncovering a study that employed the YOLOv4 model to detect and classify hip fracture types in X-ray images [34]. The findings of this study indicated that YOLOv4 achieved an accuracy of 95%, sensitivity of 96.2%, and surgeon sensitivity ranging from 69.2% to 96.2%. YOLOv4 exhibited a fracture prediction performance comparable to or better than that of a first-year resident when compared to a surgeon. However, this investigation was limited to a 2D environment as it utilized X-rays. In contrast, our study operated within a 3D environment, offering intuitive fracture extent markings for surgeons. Additionally, we came across a recent study discussing the paradigm shift in fracture detection methods through Deep Supervised Learning [35]. The study noted a significant increase in fracture detection studies utilizing artificial intelligence, with nearly 5000 conducted from 2017 to the present, marking a substantial shift from the pre-2017 period with fewer than 1000 studies. Notably, the YOLOv4 model we employed incorporates DenseNet, a feature found in only 5% of recent studies. Given DenseNet’s superior deep network performance, our model stands out from the remaining 95%. Another noteworthy discovery was a study comparing various papers focused on bone fracture detection using AI models [36]. After reviewing around 40 reliable papers, it became evident that these studies were predominantly detecting fracture sites in 2D environments, with a lack of emphasis on 3D visualization. The majority of the existing studies predominantly focused on detecting fractures in a 2D environment. In contrast, our approach stands out by automatically identifying the fracture site and providing a 3D visualization.

Although the aspects examined for the diagnosis of orthopedic trauma can vary according to the surgeons’ experiences and skill level, orthopedic surgeons can obtain assistance for a quick and clear diagnosis of orthopedic trauma regardless of their experience or skill level, by using the proposed combination of 3D reconstructed images and automatic detection of the fractured regions. Moreover, the biggest benefit of the proposed system is that it could reduce human errors, and misdiagnoses can be decreased in the clinic [37,38]. Clearly, the deep learning used in this study did not need to be trained with a lot of data. Nevertheless, the results showed the relatively good performance with an over 0.60 average precision, as seen in the PR curve in Figure 10. Naturally, the performance can be increased by training using additional data. In Figure 8 and Figure 9, some cases showed red masks located at a normal region without a fracture. This kind of error can be also dramatically decreased by additional training with more data.

This study has clear limitations in that the deep learning roughly detected the fractured region, not the individual fracture fragments. Because this study is a basic level study to show the feasibility of intuitive diagnosis support for orthopedic trauma, the use of YOLO v4 with hardware limitations and insufficient data are also other limitations of this study. Nevertheless, we found a reason why we used YOLO v4 from Table 1. YOLO v5 and v8 were not validated by the peer review paper, while the origin of YOLO v6 was YOLO v5. YOLO v7 is difficult to use in hospitals because the model requires high hardware performance. For these reasons, there are a lot of studies that have focused on YOLO v4-based object detection for medical images. For example, a research endeavor employed YOLO v4 to proficiently detect lung nodules [39]. The findings indicated the effectiveness of YOLO v4 within a hospital hardware environment. In one of the more recent investigations, object recognition using YOLO v4 was scrutinized, revealing substantial enhancements in both detection speed and accuracy. In the future, the upgraded system for the diagnosis support for orthopedic trauma can be developed with a better deep learning model and a larger dataset to detect even individual fracture fragments [40,41,42,43].

## 5. Conclusions

We confirmed the feasibility and the performance of the combination between YOLO v4-based fracture detection with intuitive visualization and 3D reconstructed bone images from the CT images. If the orthopedic surgeons use this AI-based 3D diagnosis support system for orthopedic trauma in the clinic, the surgeons can conduct easy and fast trauma diagnoses with a high accuracy regardless of their experience and skill level.

Moving forward, our strategy involves leveraging additional CT data to enhance the sophistication of our AI model. The plan is to broaden the scope of training beyond the tibia and elbow, encompassing more extensive anatomical regions. Looking ahead, our future endeavors include subjecting the implemented model to clinical trials to validate its reliability and usability in 3D visualization.

## Figures and Tables

**Figure 1 diagnostics-14-00011-f001:**
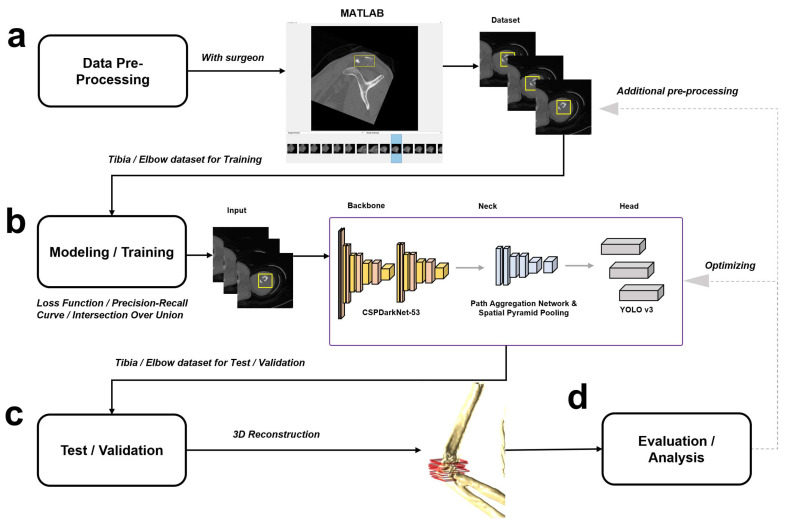
Overview of overall algorithm of the working process. (**a**) Data preprocessing was performed with the help of a surgeon using MATLAB. After upload of the computed tomography (CT) Digital Imaging and Communications in Medicine (DICOM) file including the fracture region, each bounding box (yellow box) was added according to the fracture region in the CT image. (**b**) The model was built based on YOLO v4 and was trained with a dataset made through pre-processing. (**c**) The model that has been trained is tested and validated. (**d**) The results of detection obtained by test data were evaluated by confirming the loss function, precision–recall curve, and intersection over union, and additional data preprocessing or optimization of the model was performed according to the evaluation results when it needed.

**Figure 2 diagnostics-14-00011-f002:**
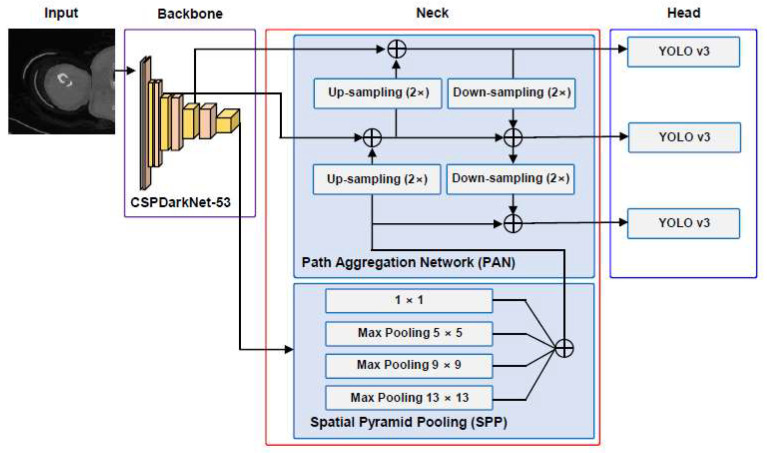
Overall structure for YOLO v4. YOLO consists of four main parts: Input, Backbone, Neck, and Head. Preprocessed data are fed into the backbone from the input. The data accepted into the backbone (purple box) are diversified through cross stage partial connection DarkNet-53 (CSPDarkNet-53). Feature maps are extracted using Spatial Pyramid Pooling (SPP) and a Path Aggregation Network (PAN) in the neck part (red box). The head part (blue box) aggregates three YOLO v3 models and generates the final output.

**Figure 3 diagnostics-14-00011-f003:**
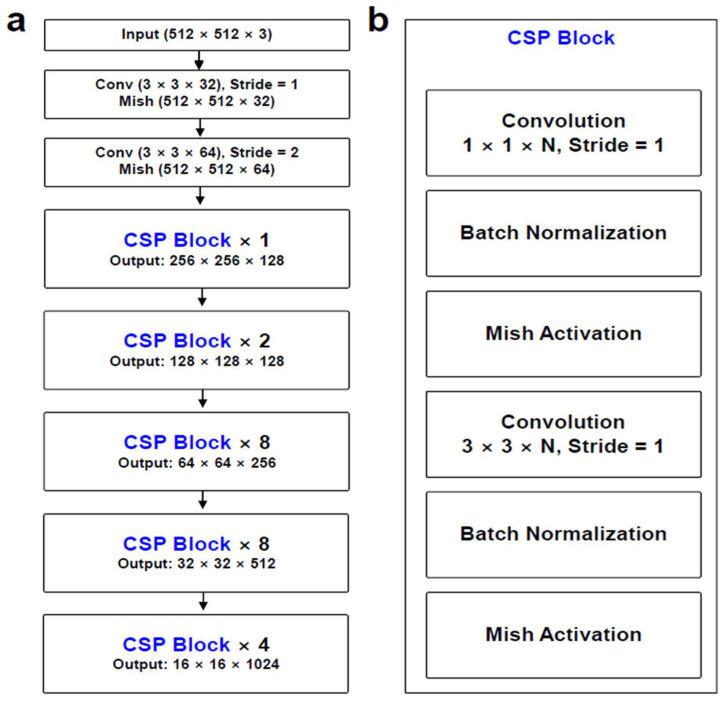
Detailed layers for backbone. Detailed layers in the overall structure of YOLO v4 (**a**) and the detailed structure for cross stage partial (CSP) blocks (**b**). In (**a**), there are multiple CSP Blocks, and each CSP Block is organized so that the size of the output can be varied. The first CSP Block has one layer and produces an output size of 256 × 256 × 128, the second CSP Block has two layers and produces an output size of 128 × 128 × 128, the third CSP Block has eight layers and produces an output size of 64 × 64 × 256, the fourth CSP Block has eight layers and produces an output size of 32 × 32 × 512, and the fifth CSP Block has four layers and produces an output size of 16 × 16 × 1024. Inside the CSP block, after convolution, batch normalization and mish activation are performed to reduce the loss value.

**Figure 4 diagnostics-14-00011-f004:**
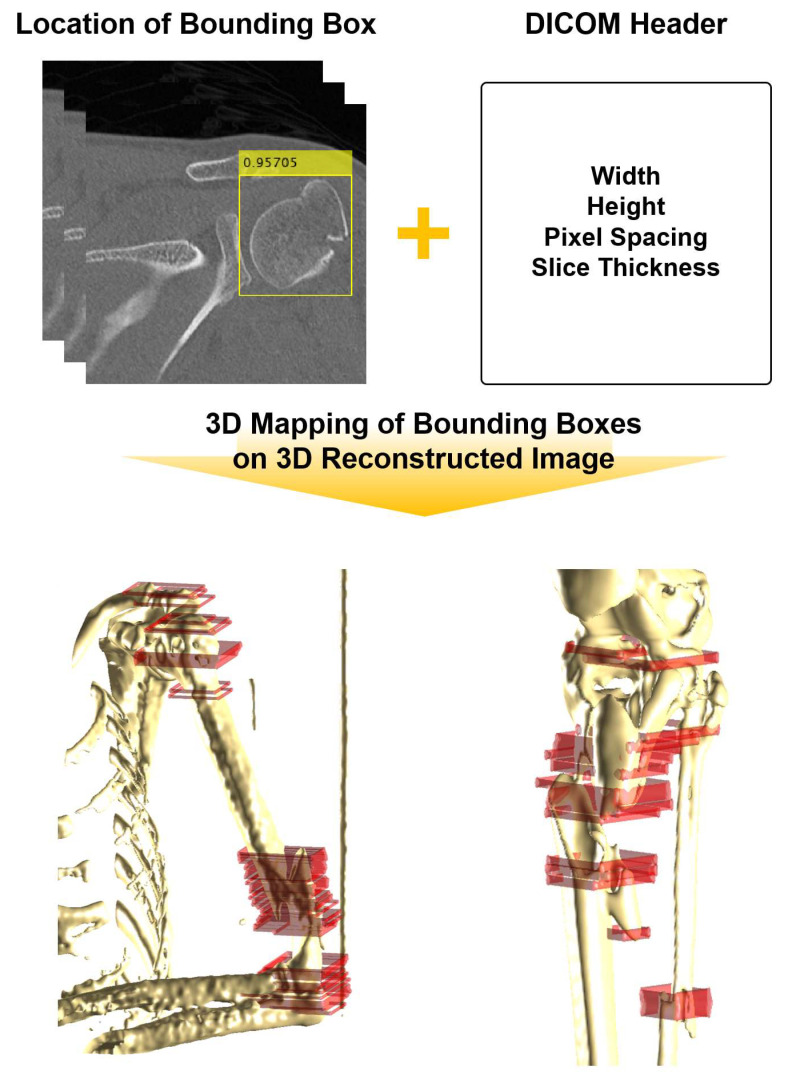
Utilizing both the bounding box’s location and size data, along with the DICOM file’s header information, the bounding box is positioned onto the 3D reconstructed bone. By converting the pixel information from the DICOM file’s header into a distance concept based on the 3D reconstructed bone, the bounding box’s placement involves calculating its width, height, and thickness from the origin axis.

**Figure 5 diagnostics-14-00011-f005:**
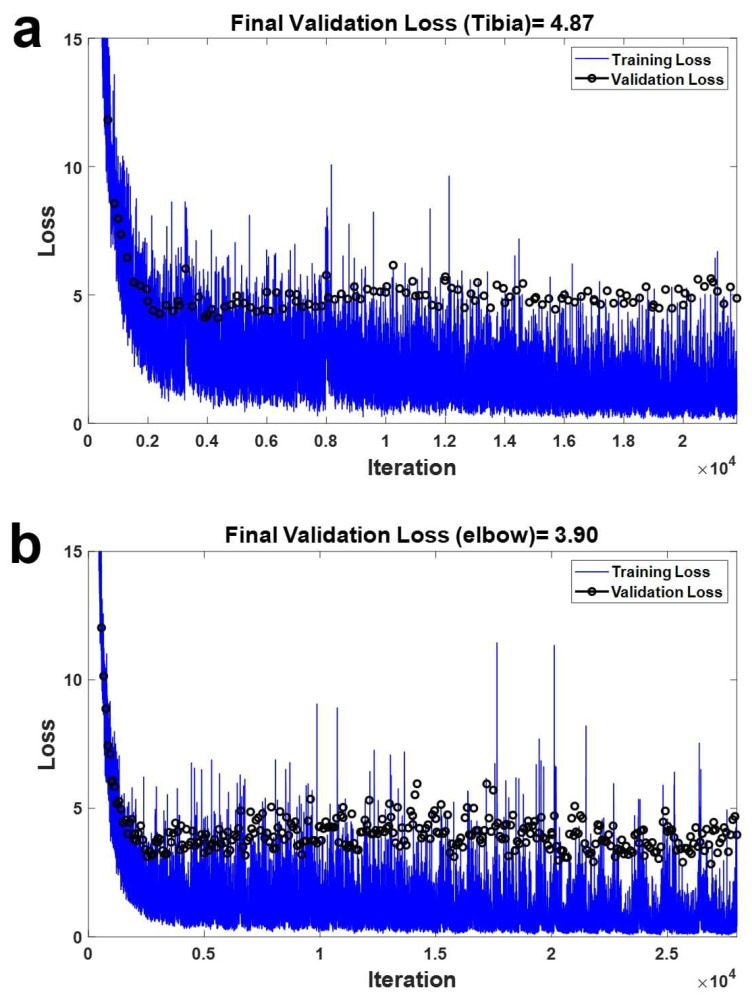
Result of validation loss after training. The Y-axis and X-axis show the loss and iteration which mean the number of validations for the minibatch. The blue line represents the variation in loss for training and the black line shows the variation in the loss in the actual validation. (**a**) Loss variation according to iteration using CT images including fracture cases in tibia region; the final validation loss was 4.87. (**b**) Variation in the loss after validation according to iteration using CT images including fracture cases in elbow region; the final validation loss was 3.90.

**Figure 6 diagnostics-14-00011-f006:**
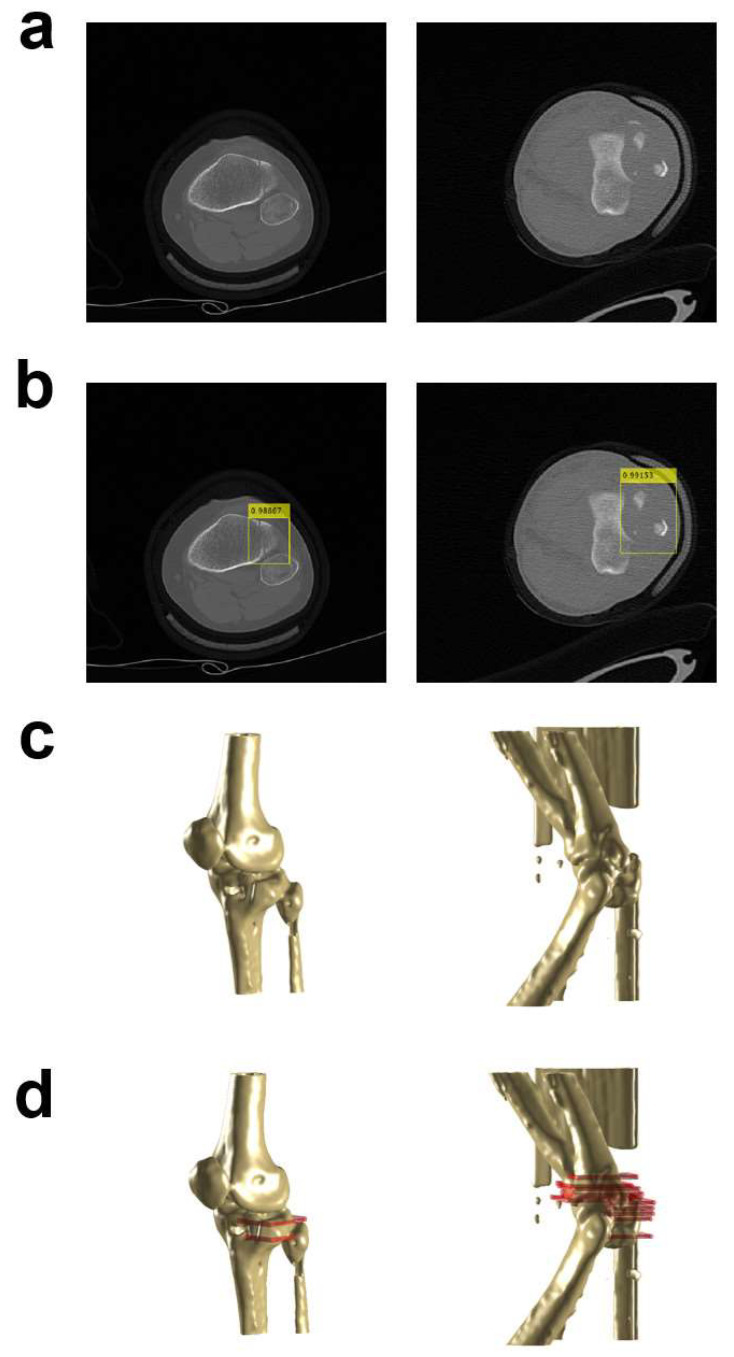
Example results for detection of fractured regions by YOLO v4 and 3D reconstructed images. The left side shows the results for the tibia case, and the right side shows the results for the elbow case. (**a**) Two dimensional (2D) images for both tibia and elbow cases including fractured regions. (**b**) Results of detection for fractured regions using images in (**a**) through YOLO v4. The yellow box is the bounding box to specify the fractured region on the image. (**c**) Three-dimensional reconstructed images using CT series including CT image in (**a**). (**d**) Three-dimensional 3D reconstructed images including red mask which can effectively show the fractured regions.

**Figure 7 diagnostics-14-00011-f007:**
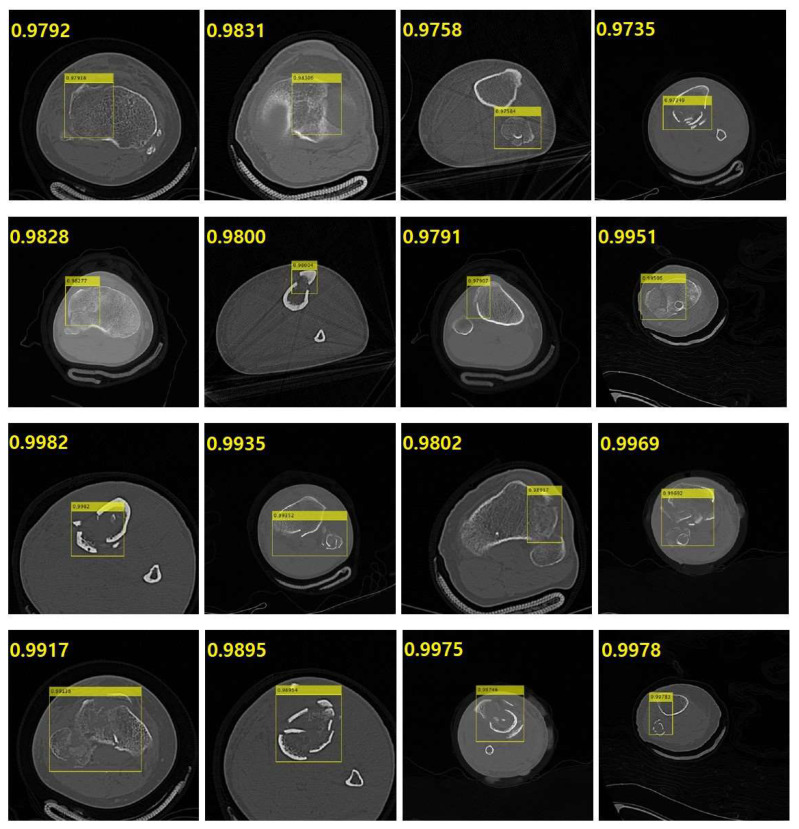
Representative results to show the detection performance for fractured region in the computed tomography (CT) images for tibia including fibula. The yellow bounding box which is generated by object detection network YOLO v4 is specifying the fractured region in the CT images. The yellow bounding box indicates not only the specific location of the fractured region but also the score which is based on the probability for a correct detection.

**Figure 8 diagnostics-14-00011-f008:**
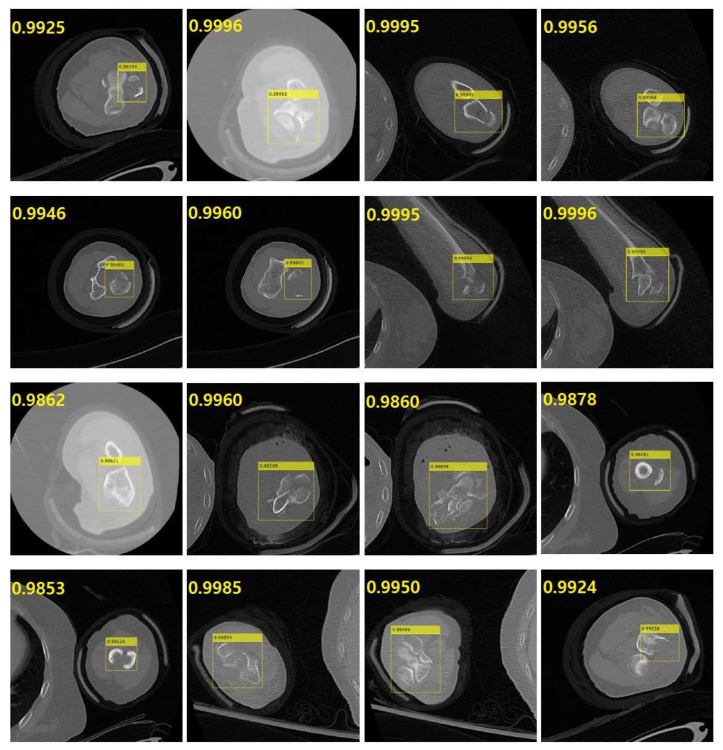
Representative results to show the detection performance for the fractured region in the computed tomography (CT) images of the elbow (humerus, radius, and ulna). The yellow bounding boxes are specifying the fractured region of the elbow in the CT images. The relatively large box is specifying the whole bone region in the CT images. In that case, the fracture type was a comminuted fracture for the whole structure which is shown in the CT image.

**Figure 9 diagnostics-14-00011-f009:**
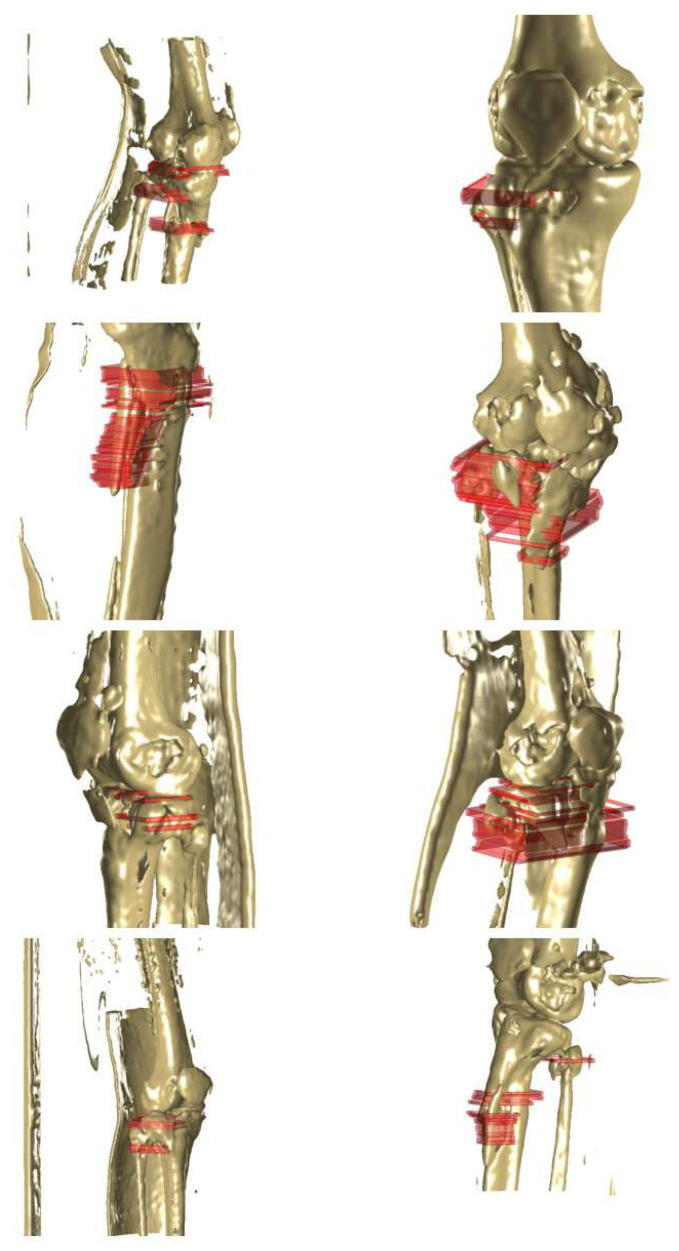
Representative results (tibia and fibula) to show the three dimensional (3D) reconstructed image for the fractured bone with red mask specifying the fractured region. The fractured bone was reconstructed as 3D images from the computed tomography (CT) series which included many CT image slides. Moreover, YOLO v4 detected the fractured region and the regions were specified by the bounding box. Although the bounding box is basically a 2D box on the individual CT image, the 3D expression (red mask) was possible by stacking the bounding boxes from several CT images in one series.

**Figure 10 diagnostics-14-00011-f010:**
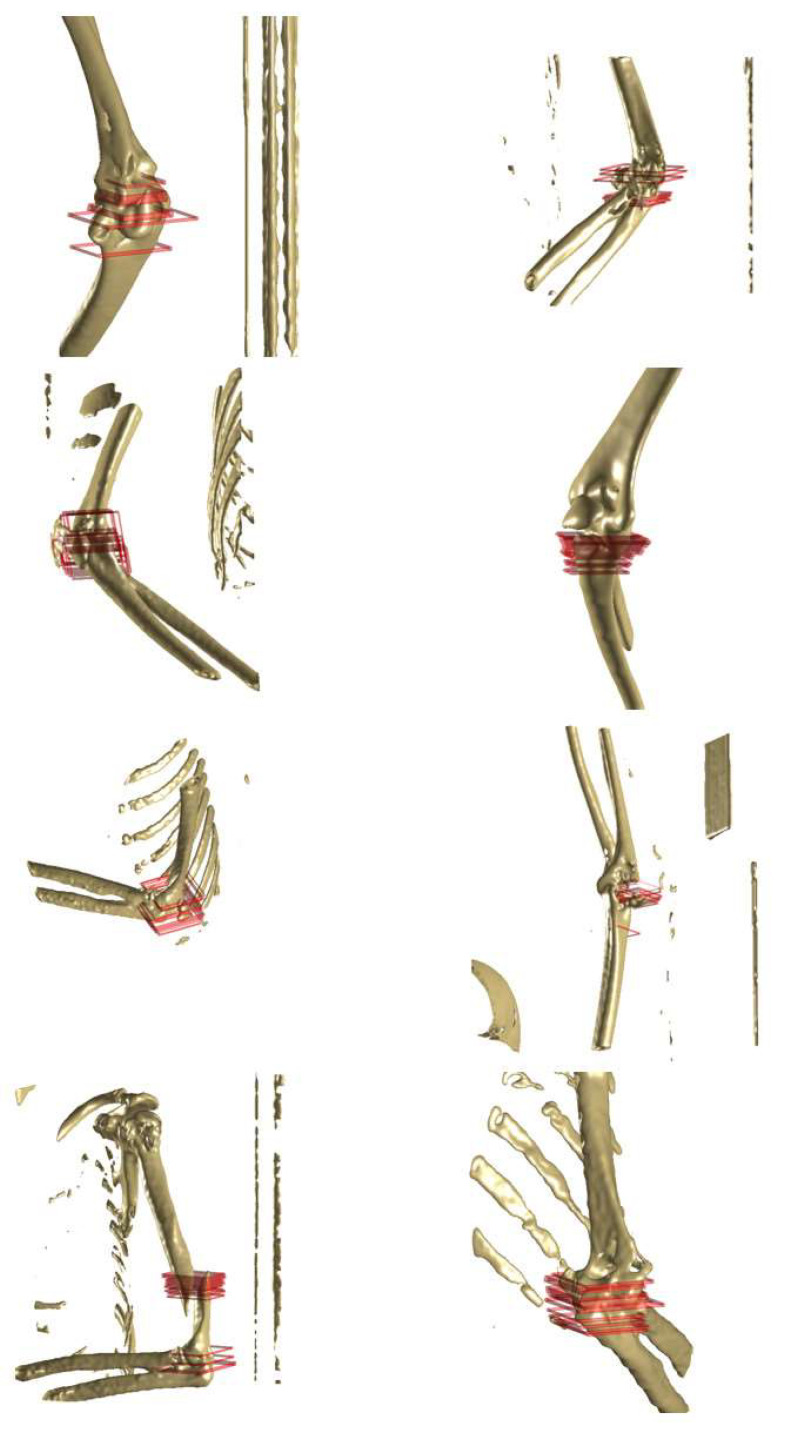
Representative results (elbow; humerus, radius, and ulna) to show the three dimensional (3D) reconstructed image for the fractured bone with a red mask specifying the fractured region. The red mask intuitively shows the fractured region in the 3D reconstructed image of the fractured elbow. The results show the multiple fractured regions using several red masks.

**Figure 11 diagnostics-14-00011-f011:**
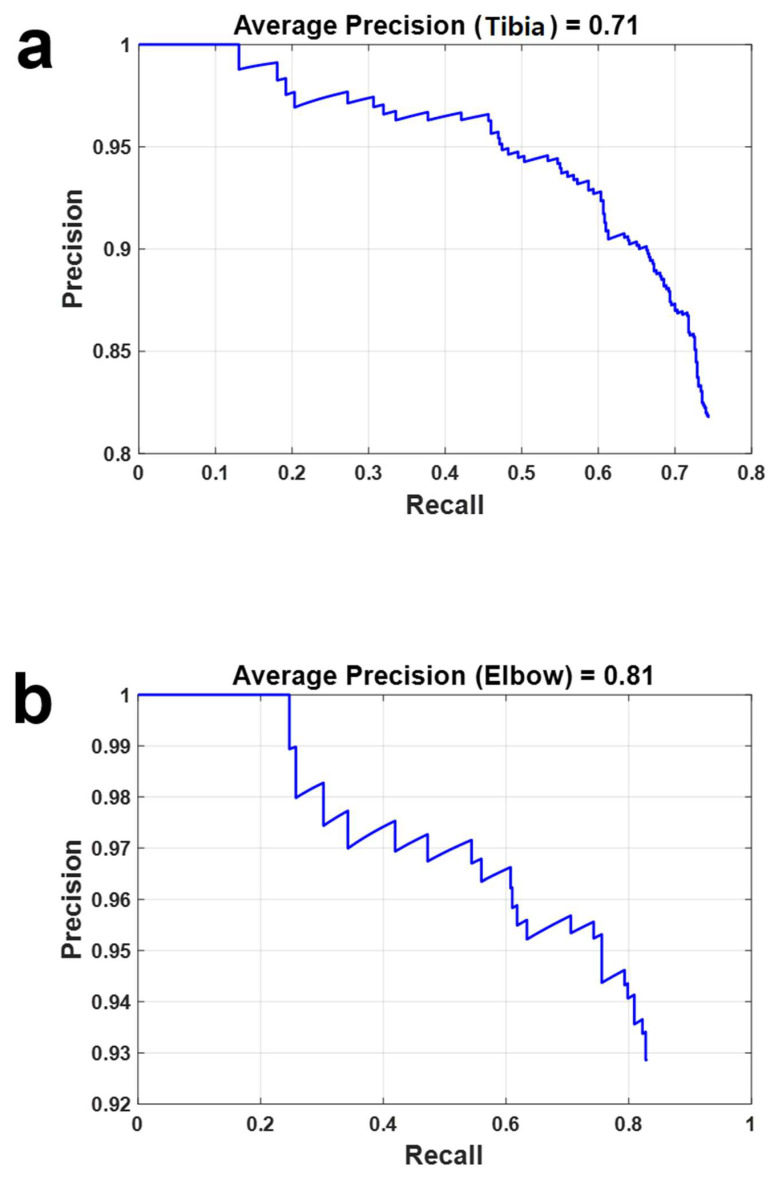
Precision–Recall (PR) curves to show the accuracy of detection of fractured regions by YOLO v4. The Y-axis and the X-axis represent the precision and the recall, respectively. The PR curves in (**a**) and (**b**) were acquired by using datasets for the tibia and elbow, respectively. The average precisions were 0.71 and 0.81 from (**a**) and (**b**), respectively.

**Table 1 diagnostics-14-00011-t001:** Comprehensive comparison table of YOLO series. Table 1 is a comparison table summarizing the YOLO v1 to v8 models’ publication date, average accuracy, whether it is a paper, and the minimum GPU performance required to run the model.

Version	Date	Paper	Average Precision (%)	Minimum Specification GPU Performance
YOLOv1	2016	YES	63.4	Very High
YOLOv2	2017	YES	63.4	Very High
YOLOv3	2018	YES	36.2	Medium
YOLOv4	2020	YES	43.5	Low
YOLOv5	2020	NO	55.8	Low
YOLOv6	2022	YES	52.5	Very High
YOLOv7	2022	YES	56.8	Medium
YOLOv8	2023	NO	53.9	Medium

## Data Availability

We have deposited the sample code we authored on GitHub (https://github.com/Louis-Youn/Code_Storage.git) to facilitate exploration by the wider community. The sample code shared via GitHub mirrors the structure of our original model. Because the full datasets are still protected by privacy issues and regulation policies, the training model can be acquired by contacting to corresponding author (D.-K.Y., E-mail: louis_youn@kavilab.ai).

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
