# Peer review of "Deep Learning Model Based on You Only Look Once Algorithm for Detection and Visualization of Fracture Areas in Three-Dimensional Skeletal Images"

_diagnostics, 2023, doi:10.3390/diagnostics14010011_

Round 1

Reviewer 1 Report

Comments and Suggestions for Authors

Review report for "Deep learning model based on YOLO for detection and visualization of fracture areas in 3D skeletal images"- Manuscript ID: diagnostics-2724455.

Technical comments:

1.      More recent related work is available. The paper should be updated to include more recent references, preferably from the last 2 or 3 years.

2.      Data augmentation was not clearly elucidated with proper explanation.

3.      Dataset was not described properly with its corresponding features.

4.      There is no proper justification for choosing MATLAB when there are so many other working platforms for object detection.

5.      Using only one evaluation metric (Precision-recall curve) is not satisfiable.

6.      3D reconstruction from 2D CT images explanation can be more detailed.

7.      The paper lacks clarity in describing the training process of the YOLO v4 model. Details on the dataset used for training, the split between training and validation sets are not provided.

8.      The use of Bag of Freebies (BoF) and Bag of Specials (BoS) in CSPDarkNet-53 has insufficient detail on how these optimization techniques contribute to the model's performance.

9.      The following paper can be cited to improve the quality of the work:

https://link.springer.com/chapter/10.1007/978-3-031-20541-5_12

10. The paper can provide an overall algorithm of the working process of the proposed work.

Overall comments:

11.  Grammatical and punctuation checks should be carried out throughout the manuscript.

12.  Figure 6. and Figure 7. Should be modified in a readable format.

13.  Highlight the major motivation and contribution of the manuscript in the introduction section.

Comments on the Quality of English Language

Minor editing of English language required

Author Response

Thank you for reviewing the manuscript. Please refer to the attached Response Letter.

<Specific Comments>

Technical comments:

1) More recent related work is available. The paper should be updated to include more recent references, preferably from the last 2 or 3 years.

[Authors’ response] Thank you for your question. I searched for recent papers and found some more suitable ones. There are still a few old references that 3-4 years ago. These are references that clearly support what I'm trying to explain. However, as you advised, I replaced most of the references with the most recent research.

[Modified reference]

  1. Ma, Wei, et al. Crop Disease Detection against Complex Background Based on Improved Atrous Spatial Pyramid Pooling. Electronics. 2023, 12.1, 216.
  2. Shardt, Yuri AW. Introduction to MATLAB®. Cham: Springer Nature Switzerland. 2023, 1-7.
  3. AboElenein, Nagwa M, et al. Encoder–Decoder Network with Depthwise Atrous Spatial Pyramid Pooling for Automatic Brain Tumor Segmentation. Neural Processing Letters. 2023, 55.2, 1697-1713.
  4. Burns, David, et al. AI in Orthopaedic Surgery. AI in Clinical Medicine: A Practical Guide for Healthcare Professionals. 2023, 266-281.
  5. Caron, Rodrigue, et al. Segmentation of trabecular bone microdamage in Xray microCT images using a two-step deep learning method. Journal of the Mechanical Behavior of Biomedical Materials. 2023, 137, 105540.
  6. Anitha, A., et al. Convolution Neural Network and Auto-encoder Hybrid Scheme for Automatic Colorization of Grayscale Images. Cham: Springer International Publishing. . 2023, 253-271.
  7. Read, Gemma JM, et al. State of science: Evolving perspectives on ‘human error’. Ergonomics. 2021, 64.9, 1091-1114.
  8. West J Emerg Med, Taiju Miyagam, et al. Medical Malpractice and Diagnostic Errors in Japanese Emergency Departments. American Journal of Medical Quality. 2023, 24.2, 340.
  9. Fishman, Elliot K, et al. 3D imaging: musculoskeletal applications. 3D Imaging in Medicine, Second Edition. 2023, 301-327.

2) Data augmentation was not clearly elucidated with proper explanation.

[Response summary] The data were augmented with a 2D affine transformation by inverting them on the x-axis or changing the size by 1-1.3 times.

[Authors’ response] Thanks for pointing that out. 2D affine transformation for both the CT images and the labeled mask was employed for the data augmentation. The affine transformation was carried out only on the X-axis. Creating data that has been inverted left and right is helpful for learning, however, data that has been inverted up and down is not helpful for learning. This is because the X-axis enhanced images can be made into datasets for both sides of the Tibia, and elbow, however, there are usually no medical images with the Y-axis inverted.  The scale was also randomly increased by 1-1.3 times. Because too large a scale change can interfere with learning, the image was allowed to increase randomly by 0–30% of the original size.

[Added contents: Page 5, Line 173 - 180] Data flowing into the backbone of the cross stage partial connection DarkNet-53 (CSPDarkNet-53) is augmented. Data augmentation primarily involved implementing 2D affine transformations on both the labeled bounding box and CT images. The images underwent X-axis flipping or random scaling transformations ranging from 1 to 1.3 times. The decision to exclude Y-axis flipping stems from the regular shape of medical images, eliminating the necessity for such transformations. In contrast, flipping along the X-axis was adopted to enhance the learning experience related to the tibia or elbow, aligning with the right and left perspectives

3) Dataset was not described properly with its corresponding features.

[Authors’ response] Thank you for your review. Added a more detailed description of the dataset. We made the dataset easy to learn by dividing it into parts for Tibia and elbow. The dataset was 60% used for learning, 30% for testing, and 10% for verification. We have written a more detailed description of how the dataset is created and used. Thanks so much for pointing out what we missed.

[Added contents: Page 4, Line 130 - 139] In the dataset construction phase, data underwent categorization based on the anatomical regions of the tibia and elbow. The categorized dataset underwent preprocessing, transforming it into a training dataset. Post MATLAB-based preprocessing for fracture site identification, the data was organized into datasets corresponding to the tibia or elbow region. Within the dataset, each entry encapsulated details about the fracture sites' location, size, and label names, assigned during the preprocessing phase. Out of the complete dataset, 60% was allocated for training, 10% for validation, and the remaining 30% for testing. Each dataset earmarked for training, validation, and testing incorporated critical information concerning the location and size of fracture labels, integral for the mapping of fracture sites onto masks in 3D reconstructed bone images.

4) There is no proper justification for choosing MATLAB when there are so many other working platforms for object detection.

[Response summary] MATLAB is reliable and has a variety of extensions, hence MATLAB has many useful points for designing models. That is why we used MATLAB to preprocess the data and design and validate the model.

[Authors’ response] Thank you for the important questions. MATLAB has additional extensions with a wide range of capabilities. They contain pre-implemented functions, hence we can quickly implement the functionality we want. The additional extensions not only help us implement them quickly, but they also come with user-friendly explanations. Also, because MATLAB has been used in research institutions for so long, it has a well-developed developer community, hence we were able to get a lot of useful feedback. Therefore, we used MATLAB because the tool has useful advantages in terms of implementation and maintenance.

[Added contents: Page 3, Line 101 - 106] MATLAB offers numerous supplementary modules tailored for researchers, providing the benefit of expediting model design. This acceleration is achieved by leveraging pre-implemented functions for preprocessing, training, validation, and testing. MATLAB holds considerable prevalence in medical research and is a robust tool, demonstrated by its application in recent drug-related studies leading to FDA approval [19].

[Added reference]

  1. Arias-Serrano, et al. Artificial intelligence based glaucoma and diabetic retinopathy detection using MATLAB—Retrained AlexNet convolutional neural network. F1000Research. 2023, 12, 14.

5) Using only one evaluation metric (Precision-recall curve) is not satisfiable.

[Response summary] The intersection over union (IoU) score means that the closer to 1, the better the performance, and the area of intersection is divided by the area of union. The IoU results showed tibia 0.6327 and elbow 0.6638 and usually if the IoU exceeds 0.5, the result means that the performance for object detection is good.

[Authors’ response] Thanks for pointing that out. As you said, evaluating the model only with the PR curve was narrow-minded. Despite the lack of learning data, the results of IoU averaged about 0.65. Tibia showed an IoU result of 0.6327, and Elbow showed an IoU result of 0.6638, indicating a compliance level of object recognition performance. We think it was very right to listen to your advice and find an evaluation method other than the PR curve. Evaluating the model using multiple evaluation methods was like looking at the model from multiple perspectives. Thank you very much.

[Added contents: Page 7, Line 244 - 250] The IoU serves as a metric indicating the success of the detection for an individual object in Object Detection. The derivation of the IoU is outlined as follows (refer to Equation (3)).

The term 'Area of Intersection' refers to the region corresponding to the correct answer, while 'Area of Union' pertains to the predicted area. The IoU achieves a maximum value of 1, with an elevated value signifying a more precise prediction[27].

[Added contents: Page 15, Line 394 - 400] In the case of the tibia, the IoU yielded a value of 0.6327, while for the elbow, the IoU registered at 0.6638. The IoU standard deviation of the calculated elbow is 0.2709 tabia, and the IoU standard deviation is 0.2754. Notably, the IoU demonstrated relatively high values despite the limited availability of learning data. Typically, when the IoU reaches 0.5 or above, the model is considered to exhibit compliant performance. This is because an IoU surpassing 0.5 implies effective coverage of the ground truth area, equivalent to two-thirds of the region[30,31].

[Added reference]

  1. Radke, Karl Ludger, et al. Adaptive IoU Thresholding for Improving Small Object Detection: A Proof-of-Concept Study of Hand Erosions Classification of Patients with Rheumatic Arthritis on X-ray Images. Diagnostics. 2023, 13.1, 104.
  2. Caron, Rodrigue, et al. Segmentation of trabecular bone microdamage in Xray microCT images using a two-step deep learning method. Journal of the Mechanical Behavior of Biomedical Materials. 2023, 137, 105540.
  3. AboElenein, Nagwa M, et al. Encoder–Decoder Network with Depthwise Atrous Spatial Pyramid Pooling for Automatic Brain Tumor Segmentation. Neural Processing Letters. 2023, 55.2, 1697-1713.

6) 3D reconstruction from 2D CT images explanation can be more detailed.

[Response summary] Based on the information about the bone in the header of the DICOM file, the 3D bone was reconstructed by constructing points and lines for the bone and filling the outer surface.

[Authors’ response] Thanks for pointing that out. We have added a clear and brief description of the process. We extracted information on area, width, slickness, and pixel for reconstructing into 3D through DICOM header information. In addition, since only image information about the bone needs to be searched, the empty matrix was filled with 1 based on the origin coordinate axis according to the Threshold value set based on the Hounsfield Unit (HU) 1150-1250, making it a matrix that extracted only the bone part. Based on the information on points, lines, and the order to connect points and lines to construct the bone surface from the completed binary matrix, the bones were constructed in a 3D space and the intact form of 3D bones was reconstructed by cap at the top and bottom.

[Added contents: Page 7, Line 253 - 275] Initially, details such as the width Initially, details such as the width and height, slice thickness, and resolution of the DICOM image were extracted from the header of the DICOM file[28,29]. Given that the 3D conversion of DICOM images necessitates exclusively retrieving bone-related image information, a blank matrix is populated with 1 based on the origin coordinate axis, aligning with the Threshold set derived from the Hounsfield Unit (HU) range of 1150-1250. Subsequently, the populated matrix is transformed into a representation isolating only the bone segments from a single DICOM file. Equivalent curves are then derived from the finalized binary matrix to construct the bone surface.

To envelop the outer epidermis of the bone, composed of dots and lines, a layer of cotton is applied, with the upper and lower sections capped to fashion a complete bone structure. For the introduction of bounding boxes onto the 3D-reconstructed bones, the DICOM image's pixel information in the DICOM header is translated into distances along the X-axis and Y-axis. Calculating the starting point of the bounding box from the origin, the bounding box is subsequently mapped from this starting point onto the X-axis, Y-axis, and thickness, strategically placed within the fracture area.

Figure 4 is a diagram that briefly shows the process of positioning the bounding box on the bone reconstructed in 3D. The DICOM header stores pixel information, height and width of the image, and threshold information of the HU range for the bone. In order to create a bounding box, pixel information on the DICOM file was converted into a 3D distance, and the starting position of the bounding box was drawn based on the coordinate axis. And the thickness of the bounding box was constructed using the information on thickness in the DICOM file. In Figure 4, red squares are bounding boxes located in the fracture area.

[Added reference]

  1. Mukherjee, Sovanlal, et al. Bounding box-based 3D AI model for user-guided volumetric segmentation of pancreatic ductal adenocarcinoma on standard-of-care CTs. Pancreatology 2023.
  2. Anitha, A., et al. Convolution Neural Network and Auto-encoder Hybrid Scheme for Automatic Colorization of Grayscale Images. Cham: Springer International Publishing. 2023, 253-271.

7) The paper lacks clarity in describing the training process of the YOLO v4 model. Details on the dataset used for training, the split between training and validation sets are not provided.

[Authors’ response] Thank you for letting us know where we missed it. Similar to question 3, but we wrote a detailed description of the learning. We have added more details to the article. The method and option used for model learning and the description of the dataset are described in more detail. We optimized learning with the Adaptive Moment Essimation (ADAM) method.  We used the dataset at a rate of 60% for training, 30% for testing, and 10% for testing.

[Added contents: Page 5, Line 166 - 171] The initial learning rate was established at 0.001, coupled with a predefined set of drop periods affecting the learning rate. The drop period, denoting the epoch interval for reducing the learning rate, was fixed at 20, accompanied by a 0.5 drop factor. Despite the initial maximum epoch being set at 50, the minibatch size was configured at 4. The chosen learning approach involved iterative training utilizing adaptive moment estimation (ADAM).

[Added contents: Page 3, Line 130 - 139] In the dataset construction phase, data underwent categorization based on the anatomical regions of the tibia and elbow. The categorized dataset underwent preprocessing, transforming it into a training dataset. Post MATLAB-based preprocessing for fracture site identification, the data was organized into datasets corresponding to the tibia or elbow region. Within the dataset, each entry encapsulated details about the fracture sites' location, size, and label names, assigned during the preprocessing phase. Out of the complete dataset, 60% was allocated for training, 10% for validation, and the remaining 30% for testing. Each dataset earmarked for training, validation, and testing incorporated critical information concerning the location and size of fracture labels, integral for the mapping of fracture sites onto masks in 3D reconstructed bone images.

8) The use of Bag of Freebies (BoF) and Bag of Specials (BoS) in CSPDarkNet-53 has insufficient detail on how these optimization techniques contribute to the model's performance.

[Response summary] Bag of Prebies (BoF) is a way to improve the performance of a model without increasing costs, and Bag of Specials (BoS) is a way to increase the performance of a model while increasing costs.

[Authors’ response] Thank you for your review. We've added the missing descriptions of Bag of Freebies (BoF) and Bag of Specials (BoS). Each technique refers to techniques that improve the performance of the model depending on whether the resource is used or not. BoF is a technique in which the performance of the model increases without increasing the cost, and BoS increases the cost, however, the performance of the model increases significantly. The difference between the two was described in more detail.

[Modified contents: Page 5, Line 194 - 204] The Bag of Freebies constitutes an amalgamation of diverse training techniques designed to enhance the accuracy of object detectors without incurring additional inference costs. This methodology represents an offline training approach aimed at elevating overall accuracy without a corresponding increase in overall inference costs[24,25]. Conversely, the Bag of Specials encompasses an assortment of plugins and post-processing modules that marginally augment the inference cost but can substantially enhance the accuracy of the object detector. The utilization of plugin modules and post-processing to slightly elevate inference costs while significantly boosting accuracy is referred to as the Bag of Specials method. Typically, these plugins are engineered to augment specific attributes, such as expanding the receptive field, introducing attention mechanisms, or fortifying feature integration capabilities.

[Added reference]

  1. Kaushik, et al. Pothole Detection System: A Real-Time Solution for Detecting Potholes. 2023 3rd Asian Conference on Innovation in Technology (ASIANCON). 2023, 1-6.
  2. Shadid, Waseem G., and Andrew Willis. Bone fragment segmentation from 3D CT imagery. Computerized Medical Imaging and Graphics. 2018, 66, 14-27.

9) The following paper can be cited to improve the quality of the work: https://link.springer.com/chapter/10.1007/978-3-031-20541-5_12.

[Authors’ response] Thanks for pointing me to a very good paper for reference. The paper is referenced on page 7, line 254. We can now further support our explanation of Figure 9. And, the paper you told me about helped me greatly to broaden my view.

[Added reference]

  1. Anitha, A., et al. Convolution Neural Network and Auto-encoder Hybrid Scheme for Automatic Colorization of Grayscale Images. Cham: Springer International Publishing. . 2023, 253-271.

10) The paper can provide an overall algorithm of the working process of the proposed work.

[Authors’ response] Thank you very much for your advice. A flowchart was created and briefly organized to make our work process easy to understand. We obtained a dataset through data preprocessing and trained the model with that dataset. Through testing and verification, the reconstructed 3D fracture site for the fracture site was evaluated.

[Added Figure: Page 4, Line 141 - 150]

Figure 1. Overview of overall algorithm of the working process. a Data preprocessing is performed with the help of a surgeon using MATLAB. After upload of the computed tomography(CT) Digital Imaging and Communications in Medicine (DICOM) file including the fracture region, each bounding box was added according to the fracture region on the CT image. b The model is built based on YOLO v4 and is learned with a dataset made through pre-processing. c The model that has been trained is tested and validated. d The results of detection obtained by test data is evaluated by confirming loss function, precision-recall curve, and intersection over union, and additional data preprocessing or optimizing the model was performed according to the evaluation results when it needed.

[Added contents: Page 3, Line 122 - 129] The model is constructed based on the YOLO v4 architecture, and training involves feeding preprocessed datasets for both the Tibia and elbow into the designed model. Following training, the model undergoes testing and validation using separately curated datasets for the Tibia and elbow. The evaluation of the 3D reconstructed bone and fracture bounding box, obtained during testing and validation, encompasses metrics such as the loss function, Precision-Recall Curve (P-R Curve), and Intersection over Union (IoU). Subsequent to this evaluation, iterations of additional data preprocessing or model optimization are conducted based on the assessment results.

Overall comments:

11) Grammatical and punctuation checks should be carried out throughout the manuscript.

[Authors’ response] Thanks for the advice. We meticulously reviewed the entire manuscript and rectified any inaccuracies or awkwardly phrased sentences comprehensively. We have tried to make it as reader-friendly as possible. Professional words were used to convey a definite meaning and the use of abbreviations was minimized.

12) Figure 6. and Figure 7. Should be modified in a readable format.

[Authors’ response] Thanks for pointing that out. As you said, I changed the value to be more prominent in the picture to make it easier to see. Values colored in yellow indicate accuracy, with values closer to 1 indicating a better prediction. The value is displayed to four decimal places.

[Modified figure: Page 11, Line 329]

[Modified figure: Page 12, Line 344]

13) Highlight the major motivation and contribution of the manuscript in the introduction section.

[Response summary] Our study is to verify the feasibility of intuitively diagnosing fractures regardless of the surgeon's skill level and combination by 3D visualization.

[Authors’ response] Thanks for the advice. we emphasized and gave meaning to why I started this study in the introduction. Our research utilizes artificial intelligence models to automatically and quickly detect fractures, making the service accessible to surgeons of all skill levels. In addition, since our system identifies the fracture site in 3D, it is possible to make an accurate diagnosis, thereby reducing the rate of misdiagnosis. This emphasizes the need for our research.

[Added contents: Page 2, Line 83 - 90] The purpose of this study is to evaluate the performance of the YOLO v4 based fracture detection, and to show the possibility of the intuitive diagnosis by combination between 3D image reconstruction and fracture detection. In 3D space, the precise location of the wound can be identified, allowing for accurate identification of fragmented or cracked bones.[14] This study is differentiated and significant in detecting fracture region more intuitively and accurately through 3D visualization[15,16]. To date, no studies have formulated a model capable of automatically reconstructing and identifying the fracture site in 3D. The imperative nature of such surgeries underscores the significance of this study

[Added reference]

  1. Fishman, Elliot K, et al. 3D imaging: musculoskeletal applications. 3D Imaging in Medicine, Second Edition. 2023, 301-327.
  2. Lavdas, Eleftherios, et al. Visualization of meniscus with 3D axial reconstructions. Journal of Medical Imaging and Radiation Sciences. 2021, 52.4, 519-526
  3. Wang, Chao, et al. Three-dimensional mapping of distal humerus fracture. Journal of Orthopaedic Surgery and Research. 2021, 16.1, 1-7.

Reviewer 2 Report

Comments and Suggestions for Authors

The study aims to investigate the feasibility of providing help to orthopedic surgeons in diagnosing fractures more accurately and intuitively using an artificial intelligence system (YOLO). This is an interesting evaluation that certainly deserves attention, some points still need to be refined:

1. In the introduction (line 68) reference is made to “a lot of studies” that show the performance of YOLO v4-base object detection in the use of medical images, it would be appropriate to mention the most important ones in order to provide a brief overview of the subject.

2. Moreover, in line 60 the Authors refer to Yolo v4 as one of the appropriate models for identifying fractures using medical images. What are the others? It would be appropriate for the Authors to describe (even briefly) what makes a fracture identification model suitable (thus underlining why yolo v4 is a good model compared to others).

3. In paragraph 2.4 the Authors deal with 3d reconstruction and plotting masks, the paragraph should be slightly expanded and explained in more detail, mostly by providing examples with images, if possible.

4. The study refers to the tibia and elbow, are other applications possible in other districts? Are there districts that may be excluded from this application.

The considerations referred to in points 1 and 2 can also be included in the discussions if necessary

Author Response

Thank you for reviewing the manuscript. Please refer to the attached Response Letter.

Reviewer 2

<General Comments>

The study aims to investigate the feasibility of providing help to orthopedic surgeons in diagnosing fractures more accurately and intuitively using an artificial intelligence system (YOLO). This is an interesting evaluation that certainly deserves attention, some points still need to be refined:

[Authors’ response] Thank you for consuming your precious time to improve the quality of our manuscript with your great comments. And we really appreciate to bring the positive aspects of our study to us. According to your comments, our manuscript has been revised. We would like to ask you to review our responses for your comments.

<Specific Comments>

1) In the introduction (line 68) reference is made to “a lot of studies” that show the performance of YOLO v4-base object detection in the use of medical images, it would be appropriate to mention the most important ones in order to provide a brief overview of the subject.

[Authors’ response] Thanks for the advice. As you suggested, we briefly described my recent research and added references. The study of finding lung nodules using YOLO v4 explains in detail the part of YOLO v4 object recognition. we can refer to the study and refer to how research on object-based detection of YOLO v4 has been conducted.

[Added contents: Page 16, Line 453 - 460] For these reasons, there are a lot of studies to show the performance of YOLO v4 based object detection using medical image. As an illustration, a research endeavor employed YOLO v4 to proficiently detect lung nodules[36]. The findings indicated the effective-ness of YOLO v4 within a hospital hardware environment. In one of the more recent investigations, object recognition using YOLO v4 was scrutinized, revealing substantial enhancements in both detection speed and accuracy. In the future, the upgraded system for the diagnosis support of the orthopedic trauma can be developed with better deep learning model and more dataset to detect even individual fracture fragment[37,38,39,40]. [Added reference]

  1. Danhui Wu, et al. An improved attention mechanism based YOLOv4 structure for lung nodule detection. 2022 IEEE Intl Conf on Dependable, Autonomic and Secure Computing, Intl Conf on Pervasive Intelligence and Computing, Intl Conf on Cloud and Big Data Computing, Intl Conf on Cyber Science and Technology Congress (DASC/PiCom/CBDCom/CyberSciTech). 2022, 1-6
  2. Rizwan. YOLOv7 and YOLOv5 Comparison on Embedded Devices and Computer Systems. 2022, https://www.plugger.ai/blog/yolov7-and-yolov5-comparison-on-embedded-devices-and-computer-systems. 38. Burns, David, et al. AI in Orthopaedic Surgery. AI in Clinical Medicine: A Practical Guide for Healthcare Professionals. 2023, 266-281.
  3. West J Emerg Med, Taiju Miyagam, et al. Medical Malpractice and Diagnostic Errors in Japanese Emergency Departments. American Journal of Medical Quality. 2023, 24.2, 340. 40. Chappard, Daniel. La microarchitecture du tissu osseux. Bulletin de l'Académie nationale de médecine. 2010, 194.8, 1469-1481.

2) Moreover, in line 60 the Authors refer to Yolo v4 as one of the appropriate models for identifying fractures using medical images. What are the others? It would be appropriate for the Authors to describe (even briefly) what makes a fracture identification model suitable (thus underlining why yolo v4 is a good model compared to others).

[Response summary] We chose YOLO v4, which satisfies the reliability, performance, and hardware requirements for the hospital environment.

[Authors’ response] Thanks for the advice. Please understand that there is an overlap with the answer to the first question while explaining only important examples. Quantitative tables were made to allow comparison with other models. The YOLO v4 we used should be available in hospital environments as well as detection performance, and it was comprehensively judged whether it was reliable. When using the YOLO model, a model is recommended to use the appropriate model according to the situation rather than the higher version unconditionally. Thanks to you, we found it very interesting to investigate the other models in more detail. Thank you very much for extending us view.

[Added contents: Page 16, Line 439 - 460] Clearly, the deep learning in this study did not be trained by using a lot of data. Nevertheless, the results showed the relatively good performance as over 0.60 average precision as the PR curve in Figure. 10. Naturally, that performance can be increased by the next training using additional data. In the Figure. 8 and 9, some cases showed the red masks which is located at normal region without the fracture. This kind of error can be also dramatically decreased by the additional training with more data.

This study has clear limitation in that the deep learning roughly detected the fractured region, not the individual fracture fragments. Because this study is a basic level study to show the feasibility of intuitive diagnosis support for the orthopedic trauma, the use of YOLO v4 with hardware limitation and insufficient data are also other limitations in this study. Nevertheless, there is a reason why we used YOLO v4. YOLO v5, v8 was not validated by the peer-review paper, the origin of YOLO v6 was YOLO v5. And YOLO v7 is difficult to use in hospitals because the model requires high hardware performance. For these reasons, there are a lot of studies to show the performance of YOLO v4 based object detection using medical image. As an illustration, a research endeavor employed YOLO v4 to proficiently detect lung nodules[36]. The findings indicated the effectiveness of YOLO v4 within a hospital hardware environment. In one of the more recent investigations, object recognition using YOLO v4 was scrutinized, revealing substantial enhancements in both detection speed and accuracy. In the future, the upgraded system for the diagnosis support of the orthopedic trauma can be developed with better deep learning model and more dataset to detect even individual fracture fragment[[37,38,39,40].

[Added reference]

  1. Danhui Wu, et al. An improved attention mechanism based YOLOv4 structure for lung nodule detection. 2022 IEEE Intl Conf on Dependable, Autonomic and Secure Computing, Intl Conf on Pervasive Intelligence and Computing, Intl Conf on Cloud and Big Data Computing, Intl Conf on Cyber Science and Technology Congress (DASC/PiCom/CBDCom/CyberSciTech). 2022, 1-6
  2. Rizwan. YOLOv7 and YOLOv5 Comparison on Embedded Devices and Computer Systems. 2022, https://www.plugger.ai/blog/yolov7-and-yolov5-comparison-on-embedded-devices-and-computer-systems. 38. Burns, David, et al. AI in Orthopaedic Surgery. AI in Clinical Medicine: A Practical Guide for Healthcare Professionals. 2023, 266-281.
  3. West J Emerg Med, Taiju Miyagam, et al. Medical Malpractice and Diagnostic Errors in Japanese Emergency Departments. American Journal of Medical Quality. 2023, 24.2, 340. 40. Chappard, Daniel. La microarchitecture du tissu osseux. Bulletin de l'Académie nationale de médecine. 2010, 194.8, 1469-1481.

3) In paragraph 2.4 the Authors deal with 3d reconstruction and plotting masks, the paragraph should be slightly expanded and explained in more detail, mostly by providing examples with images, if possible.

[Response summary] Based on the information about the bone in the header of the DICOM file, the 3D bone was reconstructed by constructing points and lines for the bone and filling the outer surface.

[Authors’ response] Thanks for the detailed advice. We have added a more detailed explanation to make it easier for readers to understand. To briefly describe the process, first, we obtained information on the width and height, slice thickness, and resolution of the DICOM image from the header of the DICOM file. Since only the image of the bone of the DICOM image should be extracted, the empty matrix was filled with 1 to fit the bone shape based on the origin coordinate axis according to the threshold value set based on the Hounsfield Unit (HU) 1150-1250. Information about the surface of the bone (equivalent curves) was extracted from the finished binary matrix. In the finished binary matrix, the outer surface was filled using the information on the outer surface of the bone and the line. The top and bottom parts were capped with caps to form a full 3D bone. To place a bounding box on a 3D reconstructed bone, the DICOM image pixel information in the DICOM header was converted to a distance relative to the X-axis and Y-axis. To place the bounding box in the fracture area, the starting point of the bounding box is calculated from the origin, and the bounding box is mapped from the starting point to the x-axis, y-axis, and thickness.

[Modified contents: Page 7, Line 253 - 275] Initially, details such as the width and height, slice thickness, and resolution of the DICOM image were extracted from the header of the DICOM file[28,29]. Given that the 3D conversion of DICOM images necessitates exclusively retrieving bone-related image information, a blank matrix is populated with 1 based on the origin coordinate axis, aligning with the Threshold set derived from the Hounsfield Unit (HU) range of 1150-1250. Subsequently, the populated matrix is transformed into a representation isolating only the bone segments from a single DICOM file. Equivalent curves are then derived from the finalized binary matrix to construct the bone surface.

To envelop the outer epidermis of the bone, composed of dots and lines, a layer of cotton is applied, with the upper and lower sections capped to fashion a complete bone structure. For the introduction of bounding boxes onto the 3D-reconstructed bones, the DICOM image's pixel information in the DICOM header is translated into distances along the X-axis and Y-axis. Calculating the starting point of the bounding box from the origin, the bounding box is subsequently mapped from this starting point onto the X-axis, Y-axis, and thickness, strategically placed within the fracture area.

Figure 4 is a diagram that briefly shows the process of positioning the bounding box on the bone reconstructed in 3D. The DICOM header stores pixel information, height and width of the image, and threshold information of the HU range for the bone. In order to create a bounding box, pixel information on the DICOM file was converted into a 3D distance, and the starting position of the bounding box was drawn based on the coordinate axis. And the thickness of the bounding box was constructed using the information on thickness in the DICOM file. In Figure 4, red squares are bounding boxes located in the fracture area.

[Added Figure: Page 4, Line 277 - 282]

Figure 5. Utilizing both the bounding box's location and size data, along with the DICOM file's header information, the bounding box is positioned onto the 3D-reconstructed bone. By converting the pixel information from the DICOM file's header into a distance concept based on the 3D-reconstructed bone, the bounding box's placement involves calculating its width, height, and thickness from the origin axis.

4) The study refers to the tibia and elbow, are other applications possible in other districts? Are there districts that may be excluded from this application.

[Authors’ response] That's a great question. We have had a lot of difficulties in collecting the data and have limitations, therefore far we have only obtained the data for Tibia and elbow. However, if we obtain more data, our model can be applied to other areas as well. If there is a part that the user wants to exclude, we can only see the part we want to see except for it. In the future, we plan to improve the performance of the model by building a dataset for more areas.

Reviewer 3 Report

Comments and Suggestions for Authors

This paper demonstrates how YOLO V4 (an instance segmentation algorithm) can detect fracture areas in skeletal images. The application chosen is impressive. The method is evaluated with a real dataset. However, the technical development is limited. Moreover, there is no comparison with the available state-of-the-art approaches for doing the same task. I find some concerns like the validation vs training loss change- it is not looking good to plot loss in 0-100 scale. I suggest a major revision to address the raised concern

Author Response

Thank you for reviewing the manuscript. Please refer to the attached Response Letter.

Reviewer 3

<General Comments>

This paper demonstrates how YOLO V4 (an instance segmentation algorithm) can detect fracture areas in skeletal images. The application chosen is impressive. The method is evaluated with a real dataset. However, the technical development is limited. Moreover, there is no comparison with the available state-of-the-art approaches for doing the same task. I find some concerns like the validation vs training loss change- it is not looking good to plot loss in 0-100 scale. I suggest a major revision to address the raised concern:

[Response summary] We also acknowledge that YOLO v4 has performance limitations. Your opinion is right. We chose YOLO v4, which satisfies the reliability, performance, and hardware requirements for the hospital environment. And our study is geared towards visualizing 3D fractures that no one has ever tried.

[Authors’ response] Thanks for pointing that out. As you said, we acknowledge that YOLO v4 has limited performance. The reason for using YOLO v4 instead of the latest version is the model chosen based on reliability, compliant performance, and relatively low hardware requirements. There are several versions of the YOLO model, however, we used the fourth version for operability and reliability reasons. YOLO v4 has validated papers and is actively being studied for use in hospitals, among other things. For example, a study was conducted using Yolo v4 to effectively detect lung nodules[11]. The study showed that Yolo v4 could work well in a hospital hardware environment. To take another YOLO model as an example, in the case of YOLO5, the reliability cannot be verified because there is no official paper and only the code has been released. YOLO v6 is derived from YOLO v5 and is considered to have high performance, however requires high-specification hardware. YOLOv7 was completed later than YOLOv6 and shows better object recognition capabilities than previous versions, however, it requires a higher hardware specification[6, 37]. In the case of YOLOv8, which is the most recent, the performance is evaluated as high, however, only the code is released without an official paper, making it difficult to verify its reliability. In other words, YOLOv4 was chosen because the model can be used in hospitals with relatively low hardware specifications, demonstrating reliability by comprehensively judging not only performance but also comprehensively. Among the studies using the YOLO series, there were studies that detected fractures or microfractures in a 2D environment, however, there were no studies that detected the fracture site and reconstructed it into 3D. Therefore, we started this study for this reason.

  The reason why the loss is relatively high when learning the model was because of the lack of data. The reason why the loss did not fall below 1 no matter how focused it was due to a lack of data. The high loss value is a phenomenon that occurs when data is insufficient in all versions as well as YOLO v4. Also, The model needs to improve learning performance through hyperparameter adjustment. However, one of the important points in this study is that the YOLO v4 model shows compliant learning performance even with little data, and focuses on the possibility of automatically reconstructing and visualizing the skeleton in 3D. We have improved the model with additional data and learning methods, and we plan to gain more data in the future to improve the performance of the model and can overcome it sufficiently. In the future, we plan to improve the performance of the model through additional data, and if we get additional data, it is a sufficiently insurmountable phenomenon.

[Added reference]

  1. Mei, et al. YOLO-lung: a practical detector based on imporved YOLOv4 for Pulmonary Nodule Detection. International Congress on Image and Signal Processing. BioMedical Engineering and Informatics. 2021, 1-6.
  2. Durve, et al. Benchmarking YOLOv5 and YOLOv7 models with DeepSORT for droplet tracking applications. The European Physical Journal E. 2023, 46.5, 32
  3. Rizwan. YOLOv7 and YOLOv5 Comparison on Embedded Devices and Computer Systems. 2022, https://www.plugger.ai/blog/yolov7-and-yolov5-comparison-on-embedded-devices-and-computer-systems.

Reviewer 4 Report

Comments and Suggestions for Authors

It seems there are several areas in the paper that require further elaboration and refinement:

  1. Clarify CSPDarkNet-53: Provide the expanded form of the abbreviation "CSP" and explain its significance in the context of the study.
  2. Detailed Methodology: Enhance the methodology section by providing more detailed information. Explain how the data was divided for training and testing. Describe the process of obtaining the ground truth for the bounding box around the fractures. Specify if the diagnosis was conducted by medical doctors (MDs).
  3. Training Details: Include specifics about the training options used, such as epoch count, minibatch size, and augmentation processes applied during training.
  4. Data and Code: Consider sharing the dataset and the code used in the study for transparency and reproducibility.
  5. Highlight Novelty: Clearly articulate the novelty of your work in the introduction or discussion sections.
  6. Comparison Table: Add a table in the discussion section to compare your study's results with existing studies for better clarity and comparison.
  7. Expand Conclusion: Elaborate on the conclusions drawn from your study and discuss potential future research directions.

Addressing these points will help bolster the clarity, depth, and completeness of your paper, enhancing its overall impact and credibility.

Author Response

Thank you for reviewing the manuscript. Please refer to the attached Response Letter.

Reviewer 4

< Specific Comments >

It seems there are several areas in the paper that require further elaboration and refinement:

1) Clarify CSPDarkNet-53: Provide the expanded form of the abbreviation "CSP" and explain its significance in the context of the study.

[Authors’ response] Thank you for your keen advice. As per your advice, the abbreviations for the cross stage partial connection DarkNet-53 (CSPDarkNet-53) were unpacked and explained, and the operation was described in more detail. Please review. Really thank you.

[Modified contents: Page 1, Line 17 - 18] The data accepted into the backbone is diversified through DarkNet-53 (CSPDarkNet-53).

[Modified contents: Page 5, Line 193 - 195] Serving as both a convolutional neural network and a backbone for object detection, CSPDarknet53 employs a split and merge strategy, promoting increased gradient flow within the network[20].

[Added reference]

  1. Wang, Chien-Yao, et al. CSPNet: A new backbone that can enhance learning capability of CNN. Proceedings of the IEEE/CVF conference on computer vision and pattern recognition workshops. 2020

2) Detailed Methodology: Enhance the methodology section by providing more detailed information. Explain how the data was divided for training and testing. Describe the process of obtaining the ground truth for the bounding box around the fractures. Specify if the diagnosis was conducted by medical surgeons (MDs).

[Response summary] With the help of a surgeon, the dataset pre-processing process was performed, and the dataset was divided into 60% for learning, 30% for testing, and 10% for verification.

[Authors’ response] Thank you so much for the detailed advice. To build a dataset to facilitate learning, we categorized the data by the tibia and elbow areas. To create training data, MATLAB was used to pre-process the fracture site, and the data were grouped into data sets corresponding to the tibia or elbow area. When setting the area in the ground truth during the pretreatment process, only the fracture area was directly labeled one by one. Ground truth is an area within a yellow bounding box. All preprocessing processes were performed with the surgeon (MD) for accurate data processing. The data in the dataset contains information about the location, size, and label name of the fracture area during the preprocessing process. Of the total dataset, 60% was used for training, 10% for validation, and the remaining 30% for testing.

[Added contents: Page 4, Line 145 - 164] The data preprocessing procedure is executed using MATLAB, with surgical intervention employed to facilitate precise preprocessing of the fracture area. The model is constructed based on the YOLO v4 architecture, and training involves feeding preprocessed datasets for both the Tibia and elbow into the designed model. Following training, the model undergoes testing and validation using separately curated datasets for the Tibia and elbow. The evaluation of the 3D reconstructed bone and fracture bounding box, obtained during testing and validation, encompasses metrics such as the loss function, Precision-Recall Curve (P-R Curve), and Intersection over Union (IoU). Sub-sequent to this evaluation, iterations of additional data preprocessing or model optimization are conducted based on the assessment results.

In the dataset construction phase, data underwent categorization based on the anatomical regions of the tibia and elbow. The categorized dataset underwent preprocessing, transforming it into a training dataset. Post MATLAB-based preprocessing for fracture site identification, the data was organized into datasets corresponding to the tibia or elbow region. Within the dataset, each entry encapsulated details about the fracture sites' location, size, and label names, assigned during the preprocessing phase. Out of the complete dataset, 60% was allocated for training, 10% for validation, and the remaining 30% for testing. Each dataset earmarked for training, validation, and testing incorporated critical information concerning the location and size of fracture labels, integral for the mapping of fracture sites onto masks in 3D reconstructed bone images

3) Training Details: Include specifics about the training options used, such as epoch count, minibatch size, and augmentation processes applied during training.

[Authors’ response] Thanks for the advice. we've added more details to the article. The method and option used for model learning and the description of the dataset are described in more detail. We performed iterative learning using the adaptive moment estimation (ADAM) technique to learn the model effectively.

[Added contents: Page 5, Line 179 - 184] The initial learning rate was established at 0.001, coupled with a predefined set of drop periods affecting the learning rate. The drop period, denoting the epoch interval for reducing the learning rate, was fixed at 20, accompanied by a 0.5 drop factor. Despite the initial maximum epoch being set at 50, the minibatch size was configured at 4. The chosen learning approach involved iterative training utilizing adaptive moment estimation (ADAM).

4) Data and Code: Consider sharing the dataset and the code used in the study for transparency and reproducibility.

[Response summary] Anyone can experience the learning ability of the YOLO v4-based model we designed through the sample code. The code was published on GIT (https://github.com/back582/master-thesis/releases/tag/data).

[Authors’ response] Thank you for your advice. we registered the sample code we wrote with GIT(https://github.com/back582/master-thesis/releases/tag/data) so that you can try it out. Please note that our sample code shared with GIT was written in exactly the same way as the model we wrote, however, the code was modified to be reproduced as a sample elbow dataset. The data preprocessed from the sample code was selected only by entering the input and labeling it as a bounding box, and a dataset was augmentation with 90% of learning data and 10% of test data. The size of the input image was set to 512×512. Resize the images of preprocessed data and scale the pixels to between 0 and 1, and also scale the corresponding bounding boxes. The elbow data were data augmentation by inverting the X-axis with a 2D affine transformation and changing the size 1-1.3 times random. The constructed dataset was used for learning and testing through deep learning network layers using K-fold cross-validation, the partition randomly divides the observations into 5 disjoint subsamples.

[Added contents: Page 18, Line 502 - 505] We have deposited the sample code we authored on GIT (https://github.com/back582/master-thesis/releases/tag/data) to facilitate exploration by the wider community. The sample code shared via GIT mirrors the structure of our original model, albeit with modifications to enable its replication as an illustrative elbow dataset.

5) Highlight Novelty: Clearly articulate the novelty of your work in the introduction or discussion sections.

[Response summary] An important point of our study is to verify the possibility of intuitive diagnosis by visualizing the fracture site in 3D using YOLO v4.

[Authors’ response] Thanks for the advice. we emphasized and gave meaning to why we started this study in the introduction. Regardless of the surgeon's skill level and experience, using our system can help a lot to make an intuitive diagnosis. Above all, because it visualizes the fracture area in 3D, it can save time and reduce the misdiagnosis rate with intuitive diagnosis. This emphasizes the need for our research.

[Added contents: Page 2, Line 66 - 91] As an illustration, a research endeavor employed YOLO v4 to proficiently detect lung nodules[11]. The findings indicated the effectiveness of YOLO v4 within a hospital hardware environment. In one of the more recent investigations, object recognition using YOLO v4 was scrutinized, revealing substantial enhancements in both detection speed and accuracy. Notably, a recent study harnessed YOLO v4 to explore the classification and detection capabilities of extensive medical data, demonstrating commendable performance[12]. The ongoing refinement of YOLO v4's performance is evident, with continual updates and user-driven feedback contributing to its improvement. Grounded in these investigations, it is discerned that YOLO v4's performance has experienced consistent advancement, underscoring the ongoing research endeavors in the realm of object-based detection. And, the YOLO v4 is one of proper models to detect bone fracture using medical image[13]. When the YOLO v4 detected the fractured region using bounding box on the 2D computed to demography (CT) image one slide by one slide, the several bounding boxes along the several image slides can show intuitive fractured region with the 3D reconstructed image for the bone region. In that case, more accurate diagnosis can be progressed, the error rate in diagnosis will be innovatively reduced by the surgeons before the operation. Moreover, leveraging YOLO v4, the system can intuitively guide diagnoses, irrespective of the surgeon's skill level and experience, further enhancing its efficacy[10]. The purpose of this study is to evaluate the performance of the YOLO v4 based fracture detection, and to show the possibility of the intuitive diagnosis by combination between 3D image reconstruction and fracture detection. In 3D space, the precise location of the wound can be identified, allowing for accurate identification of fragmented or cracked bones.[14] This study is differentiated and significant in detecting fracture region more intuitively and accurately through 3D visualization[15,16]. To date, no studies have formulated a model capable of automatically reconstructing and identifying the fracture site in 3D. The imperative nature of such surgeries underscores the significance of this study.

[Added reference]

  1. Mei, et al. YOLO-lung: a practical detector based on imporved YOLOv4 for Pulmonary Nodule Detection. International Congress on Image and Signal Processing. BioMedical Engineering and Informatics. 2021, 1-6.
  2. Wu, et al. Small-target weed-detection model based on YOLO-V4 with improved backbone and neck structures. Precision Agriculture. 2023, 1-22.
  3. Bochkovskiy, Alexey, Chien-Yao Wang, and Hong-Yuan Mark Liao. Yolov4: Optimal speed and accuracy of object detection. arXiv preprint arXiv. 2020, 2004, 10934.
  4. Fishman, Elliot K, et al. 3D imaging: musculoskeletal applications. 3D Imaging in Medicine, Second Edition. 2023, 301-327.
  5. Lavdas, Eleftherios, et al. Visualization of meniscus with 3D axial reconstructions. Journal of Medical Imaging and Radiation Sciences. 2021, 52.4, 519-526
  6. Wang, Chao, et al. Three-dimensional mapping of distal humerus fracture. Journal of Orthopaedic Surgery and Research. 2021, 16.1, 1-7.

6) Comparison Table: Add a table in the discussion section to compare your study's results with existing studies for better clarity and comparison.

[Response summary] We chose YOLO v4, which satisfies the reliability, performance, and hardware requirements for the hospital environment.

[Authors’ response] Thanks for bringing up a very perceptive point. You have expanded my view. We've tried to make the comparison as reliable as possible. We also need to compare the overall performance because we need to consider not only the accuracy of the AI, but also the ability to run on low-end hardware in hospitals.

[Added table: Page 16, Line 430 - 438] Additionally, it is imperative to conduct more nuanced comparative studies to determine the necessity of utilizing YOLO v4. Despite the proliferation of YOLO model versions, YOLO v4 demonstrated notable efficacy and accuracy even in resource-constrained environments. Noteworthy is the observation that YOLO v4 not only exhibited compliant performance but also gained credibility due to the availability of relevant research papers. When opting for a model, careful consideration should be given to selecting the most suitable one based on its accuracy and the specific context of its application. The accompanying table provides a reference for the requisite conditions related to performance and utilization.

[Added Table: Page17, Line 468 - 471]

Table 1. Comprehensive comparison table by YOLO series. Table 1. is a comparison table summarizing the model's publication date for YOLO v1 to v8, the average accuracy, whether it is a paper, and the minimum GPU performance required to run the model.

Version

Date

Paper

Average Precision

(%)

Minimum Specification GPU performance

YOLOv1

2016

YES

63.4

Very High

YOLOv2

2017

YES

63.4

Very High

YOLOv3

2018

YES

36.2

Medium

YOLOv4

2020

YES

43.5

Low

YOLOv5

2020

NO

55.8

Low

YOLOv6

2022

YES

52.5

Very High

YOLOv7

2022

YES

56.8

Medium

YOLOv8

2023

NO

53.9

Medium

[Added reference]

  1. Juan R, Diana M. A comprehensive review of YOLO: From YOLOv1 and beyond. arXiv 2023. arXiv preprint arXiv. 2023, 2304, 00501.
  2. Shuo Wang. Research Towards Yolo-Series Algorithms: Comparison and Analysis of Object Detection Models for Real-Time UAV Applications. Journal of Physics: Conference Series. 2021, 1, 012021.

Round 2

Reviewer 3 Report

Comments and Suggestions for Authors

In the previous review, I drew serious concern about the availability of methodical development. However, the authors treated my comments as a performance limitation. The rest of the comments are also not addressed.

Author Response

Thank you for reviewing the manuscript. Please refer to the attached Response Letter.

Reviewer 3

<General Comments>

In the previous review, I drew serious concern about the availability of methodical development. However, the authors treated my comments as a performance limitation. The rest of the comments are also not addressed.

[Response summary] We explained why we chose the method by comparing it to the latest methods, and modified validation versus training loss changes. Thanks to you, we were able to plan to complement the weaknesses of our system. Thank you very much for your feedback.

[Authors’ response] Thank you very much for giving us feedback once again. The whole process we went through together grew us up and we were able to learn your philosophy and thoughts. We recognized that we lacked an explanation for the methodology and lacked an explanation for the training vs verification loss. You are comment is also true that there is a limit to the availability of development that you have argued for. We did our best to answer your question.

  1. Compared to the latest fracture diagnostic technology

We explored innovative approaches to fracture diagnosis and came across a study utilizing the YOLOv4 model to identify and categorize hip fracture types in X-ray images[1]. The YOLOv4 demonstrated an accuracy of 95% and a sensitivity of 96.2%. When compared to a surgeon, YOLOv4 exhibited similar or superior fracture prediction performance to that of a first-year resident, despite the study being confined to a 2D environment due to the use of X-rays. In contrast, our study involved 3D visualization and reconstruction, automatically delineating the fracture extent for intuitive diagnosis by surgeons. Additionally, we encountered a recent study addressing the paradigm shift in fracture detection methods through Deep Supervised Learning[2]. Before 2017, there were fewer than 1,000 studies related to artificial intelligence models for detection, but the trend shifted, with nearly 5,000 studies conducted from 2017 to the present. Notably, the YOLOv4 we employed incorporated DenseNet, a model utilized in only 5% of recent studies. Given DenseNet's advanced deep network performance, our model stands out among the remaining 95% of models. Lastly, we came across a study comparing various papers on bone fracture detection through AI models[3]. After scrutinizing around 40 reliable papers, we observed a common focus on detecting fracture sites in 2D environments through AI. Interestingly, no studies were found that visualized the detected sites in 3D. Our system distinguishes itself from these contemporary technologies by offering automatic fracture site detection and 3D visualization.

Our method has limitations in availability like your feedback, however, this is a problem that can be sufficiently overcome through data. Problems arising from lack of data are common to all AI, not only YOLOv4. We took your comment very seriously. We have thought about what's better about our method, and how our method can develop further in the future. We are preparing to secure more and more diverse data to solve the problem of lack of learning data. The model that applied YOLO v4 we developed showed relatively good learning with little data, however, still did not completely solve the problem caused by the lack of data. Therefore, we are seeking data on more diverse fracture patterns to recognize a wider range of fracture sites and are also planning to collect additional data on previously learned tibia, and elbow. We plan to conduct future clinical trials of our system and verify the method's ability to intuitively find fracture sites in a 3D environment.

  1. Changed confirmation versus training loss change graph.

The alteration in our validation versus training loss representation did not yield a favorable outcome, and we acknowledge your observation. We initially opted for a scale from 0 to 100 to illustrate that YOLO exhibits commendable learning performance, even with a limited dataset.  We have modified the Final validation loss Tibia/elbow graph by eliminating the superfluous Y-axis and highlighting the pertinent segment. In response to your feedback, we have adjusted the scale to reflect a range of 0 to 30. Figure 5 has changed intuitively to check the amount of change than before.

[Added contents: Page 16, Line 394 – 414] We explored the latest methodologies for diagnosing fractures, uncovering a study that employed the YOLOv4 model to detect and classify hip fracture types in X-ray images[34]. The findings of this study indicated that YOLOv4 achieved an accuracy of 95%, sensitivity of 96.2%, and surgeon sensitivity ranging from 69.2% to 96.2%. YOLOv4 exhibited fracture prediction performance comparable to or better than that of a first-year resident when compared to a surgeon. However, this investigation was limited to a 2D environment as it utilized X-rays. In contrast, our study operates within a 3D environment, offering intuitive fracture extent markings for surgeons. Additionally, we came across a recent study discussing the paradigm shift in fracture detection methods through Deep Supervised Learning[35]. The study noted a significant increase in fracture detection studies utilizing artificial intelligence, with nearly 5,000 conducted from 2017 to the present, marking a substantial shift from the pre-2017 period with fewer than 1,000 studies. Notably, the YOLOv4 model we employed incorporates DenseNet, a feature found in only 5% of recent studies. Given DenseNet's superior deep network performance, our model stands out from the remaining 95%. Another noteworthy discovery was a study comparing various papers focused on bone fracture detection through AI models[36]. After reviewing around 40 reliable papers, it became evident that these studies were predominantly detecting fracture sites in 2D environments, with a lack of emphasis on 3D visualization. The majority of existing studies predominantly focused on detecting fractures in a 2D environment. In contrast, our approach stands out by automatically identifying the fracture site and providing a 3D visualization.

[Modified figure Page: 9, Line: 276]

[reference]

  1. Twinprai, et al. Artificial intelligence (AI) vs. human in hip fracture detection. Heliyon. 2022, 8.11.
  2. Meena, et al. Bone fracture detection using deep supervised learning from radiological images: A paradigm shift. Diagnostics. 2022, 12.10, 2420.
  3. Su, et al. Skeletal Fracture Detection with Deep Learning: A Comprehensive Review. Diagnostics. 2023, 13.20, 3245.

Reviewer 4 Report

Comments and Suggestions for Authors

The paper much improved 

Author Response

Thank you for reviewing the manuscript. Please refer to the attached Response Letter.

Reviewer 4

[Authors’ response] Thanks to you, our research was able to move in a better direction. Your question was sharp and was the feedback we needed. The whole process we went through together grew us up and we were able to learn your philosophy and thoughts. Once again, thank you very much for giving us a good opportunity. We hope you have a good day.

Many thanks to you.

-Authors-

Round 3

Reviewer 3 Report

Comments and Suggestions for Authors

No comments.